# GENERALIZING TO ANY DIVERSE DISTRIBUTION: UNIFORMITY, GENTLE FINETUNING & REBALANCING

## ABSTRACT

As training datasets grow larger, we aspire to develop models that generalize well to any diverse test distribution, even if the latter deviates significantly from the training data. Various approaches like domain adaptation, domain generalization, and robust optimization attempt to address the out-of-distribution challenge by posing assumptions about the relation between training and test distribution. Differently, we adopt a more conservative perspective by accounting for the worst-case error across all sufficiently diverse test distributions within a known domain. Our first finding is that training on a uniform distribution over this domain is optimal. We also interrogate practical remedies when uniform samples are unavailable by considering methods for mitigating non-uniformity through finetuning and rebalancing. Our theory aligns with previous observations on the role of entropy and rebalancing for o.o.d. generalization and foundation model training. We also provide new empirical evidence across tasks involving o.o.d. shifts which illustrate the broad applicability of our perspective.

## 1 INTRODUCTION

Machine learning usually starts with the assumption that the test data will be independent and identically distributed (i.i.d.) with the training set. In practice, distributional shifts are the norm rather than the exception, leading to models that perform well in training but may stumble when faced with the diversity the real world has to offer.

The challenge of out-of-distribution (o.o.d.) generalization has inspired a variety of approaches aimed at bridging the training and inference gap. For example, approaches like domain adaptation and generalization address o.o.d. challenges by assuming knowledge of the unlabeled test distribution or by learning invariant features (Bengio et al., 2013; Peters et al., 2016; Arjovsky et al., 2019; Rosenfeld et al., 2020; Koyama & Yamaguchi, 2020), whereas robust optimization (Ben-Tal et al., 2009; Rahimian & Mehrotra, 2019) methods can be used to defend against data uncertainty by modifying and regularizing training.

We take a different perspective and seek models that perform well under *any* diverse test distribution within a known domain. This is formalized through the concept of *distributionally diverse (DD) risk*, which quantifies the worst-case error across all distributions with sufficiently high entropy. Our postulate that test entropy is large reflects our intention to characterize a model's performance on a sufficiently diverse set of natural inputs rather on adversarial examples. Our focus on entropy is also motivated by the previous empirical finding that higher entropy in training and test data is a strong predictor of o.o.d. generalization (Vedantam et al., 2021).

The introduced framework provides a new angle to study o.o.d. generalization. Differently from domain generalization, we do not assume that the training data are composed of multiple domains. Further, unlike domain adaptation and robust optimization, we do not assume to know the unlabeled test distribution nor that the latter lies close to the training distribution. A more comprehensive discussion of how our ideas relate to previous work can be found in Appendix A.

Our analysis starts in Section 3 by showing that DD risk minimization has a remarkably simple solution when we have control over how training data are sampled. Specifically, we prove that training on the uniform distribution over the domain of interest is optimal in the worst-case scenario and derive a matching bound on the corresponding DD risk.

Section 4 then explores what happens when the training data is non-uniformly distributed. We analyze two approaches. First, we show that gentle finetuning of a pretrained model (as opposed to considerably deviating from the pre-training initialization) can suffice to overcome the non-uniformity issue. Second, we draw inspiration from test-time adaptation and formally consider the re-weighting of training examples to correct distributional imbalances. Therein, we provide an end-to-end generalization bound that jointly captures the trade-off that training set rebalancing introduces between in- and out-of-distribution error.

The above results agree with past observations in the training of large models. Our analysis of finetuning provides an explanation for the observation of Chen et al. (2024) (specifically Figure 4) that when finetuning large language models to respect human preferences, i.i.d. and o.o.d. metrics correlate only close to finetuning initialization. Further, our findings on the role of uniformity and on benefit of rebalancing align with emerging empirical observation within foundation model training, such as large language models (Gao et al., 2020; Furuta et al., 2023; Dai et al., 2024) and AlphaFold (Jumper et al., 2021; Abramson et al., 2024), where heuristic ways to do training set rebalancing, such as clustering or controlling the data source mixture, are adopted to improve generalization and to remove bias.

Our insights are further evaluated in syntetic and real world tasks featuring distribution shifts. After first validating the theory in a controlled and tractable setup involving a mixture of Gaussian distributions, we turn our attention to complex tasks involving covariate shift. Therein, we find that rebalancing can enhance empirical risk minimization when the density can be reasonably estimated. These results exemplify the practical benefits and pitfalls of the considered approaches.

## 2 DISTRIBUTIONALLY DIVERSE RISK

A holy grail of supervised learning is to identify a function that minimizes the worst-case risk

$$r_{\mathrm{wc}}(f) = \max_{x \in \mathcal{X}} l\big(f(x), f^*(x)\big),$$ (1)

where $l$ is a loss function, such as the zero-one or cross-entropy loss for classification, $x \in \mathcal{X}$ are examples from some domain of interest, and $f^*(x)$ is some unknown target function. Unfortunately, it is straightforward to deduce that minimizing the worst-case risk by learning from observations is generally impossible unless one is given every possible input-output pair.

The usual way around the impossibility of worst-case learning involves accepting some probability of error w.r.t. a distribution $p$. The expected risk is defined as

$$r_{\mathrm{exp}}(f; p) = \mathbb{E}_{x \sim p}\Big[l\big(f(x), f^*(x)\big)\Big].$$ (2)

The above definition is beneficial because it allows us to tractably estimate the error of our model using a validation set or a mathematical bound. However, the obtained guarantees are limited to i.i.d. examples from $p$, and the model's predictions can be due to spurious correlations and entirely unpredictable, otherwise. The focus of this work is to propose an alternative requirement that bridges the gap between the worst- and average-case perspectives.

We instead look for models whose average-case error under *any* sufficiently diverse distribution within a domain $\mathcal{X}$ is bounded. Concretely, consider a compact domain of interest $\mathcal{X}$ that contains the test data under consideration as a subset with sufficiently high probability, i.e., $q(\mathcal{X}) \approx 1$ for any test distribution $q$. In the small and medium data regimes, the domain $\mathcal{X}$ should be defined by prior knowledge about the task in question. In the large data regime, such as when training foundation models, we may consider $\mathcal{X}$ as the set of all natural objects. Further, let $\mathcal{Q}_\gamma$ be the set of distributions $q$ supported on $\mathcal{X}$ with entropy $H(q) \geq H(u) - \gamma$, where $H(u) = \log(\mathrm{vol}(\mathcal{X}))$ is the entropy of the uniform distribution on $\mathcal{X}$ expressed in 'nat' (log indicates the natural logarithm). We define the distributionally diverse (DD) risk as follows:

$$r_{\mathrm{dd}}(f; \gamma) = \max_{q \in \mathcal{Q}_\gamma} \mathbb{E}_{x \sim q}\Big[l\big(f(x), f^*(x)\big)\Big].$$ (3)

In simple terms, DD risk seeks to measure performance across a broad range of diverse distributions, rather than a single, known distribution. Further justification can be found in Appendix B). We

emphasize that the DD risk focuses on covariate shift but not other types of distribution shift, such as label and concept drift.

DD risk subsumes the worst-case risk as a special case,

$$\lim_{\gamma \to \infty} r_{\mathrm{dd}}(f; \gamma) = r_{\mathrm{wc}}(f), \tag{4}$$

which follows from that, as $\gamma$ increases, $\mathcal{Q}_\gamma$ contains distributions all of whose mass lies arbitrarily close to the point of maximal loss. Though it is easy to also deduce that the DD risk is always larger than the expected risk, it turns out that the gap between the two can be zero when considering the optimal learner. We refer to Appendix D for a more in depth analysis of this theoretical topic and turn our attention to more practical matters.

## 3 UNIFORM IS OPTIMAL AND IMPLIED GUARANTEES

This section argues that –all other things being equal– it is preferable in terms of distributionally diverse risk to train your classifier on the uniform distribution. We then characterize the DD risk of a classifier that achieves a certain expected risk on the uniform distribution.

### 3.1 LEARNING FROM A UNIFORM DISTRIBUTION IS DD RISK OPTIMAL

Suppose that there exists some unknown function $f^* : \mathcal{X} \to \mathcal{Y}$ and that the learning algorithm determines a classifier $f : \mathcal{X} \to \mathcal{Y}$ whose expected risk with respect to a distribution $p$ is $\varepsilon$.

Denote by $\mathcal{F}_{p,\varepsilon}$ the set of all classification functions that the learner may have selected:

$$\mathcal{F}_{p,\varepsilon} := \{f : \mathcal{X} \to \mathcal{Y} \text{ such that } \mathbb{E}_{x \sim p}[\ell(f(x), f^*(x))] = \varepsilon\}. \tag{5}$$

In the following theorem, we consider how the choice of the training distribution $p$ affects the worst-case DD risk within $\mathcal{F}_{p,\varepsilon}$.

**Theorem 3.1.** *Consider a zero-one loss and suppose that we can train a classifier up to some fixed expected risk $\varepsilon < 1/2$ under any distribution. A classifier optimized for the uniform distribution will yield the smallest DD risk:*

$$\max_{f \in \mathcal{F}_{u,\varepsilon}} r_{dd}(f; \gamma) \leq \max_{f \in \mathcal{F}_{p,\varepsilon}} r_{dd}(f; \gamma) \quad \text{for all} \quad p \neq u. \tag{6}$$

The proof can be found in Appendix E. Intuitively, a uniform distribution is optimal because it balances the model's performance across the entire input space, preventing overemphasis of specific areas. The reader might suspect that this result is a consequence of the maximum entropy principle, stating that within a bounded domain the uniform distribution has the maximum entropy. This is indeed accurate, although the derivation is not a straightforward application of this result: the maximum entropy principle constrains the choice of the worst-case distribution $q^*$ within $\mathcal{Q}_\gamma$.

A particularly appealing consequence of the theorem is that the exact entropy gap $\gamma$ is not necessary to determine the optimal training strategy. As we shall see later, $\gamma$ does affect the DD risk that we can expect. However, from a practical perspective it is preferable to have a training strategy that is independent of $\gamma$, as it may not be straightforward to define it.

It is also important to discuss when a uniform distribution is not the optimal choice for training. Our first disclaimer is that the theorem does not account for any inductive bias in learning, e.g., as afforded by the choice of data representation, model type, and optimization. The theorem also does not consider the pervasive issue of i.i.d. generalization, meaning how close the empirical risk approximates the expected one, which is analysed in Section 4.3. Finally, the theorem is less relevant when there is additional information about the test distribution, such as unlabeled test samples.

### 3.2 THE GAP BETWEEN EXPECTED AND DISTRIBUTIONALLY DIVERSE RISK

Our next step entails characterizing the relation between distribution DD risk and expected risk. We will show that the DD risk can be upper bounded by the expected risk with respect to the uniform

distribution, implying that expected risk minimization with a uniform distribution is a good surrogate for DD risk minimization.

Supposing we know the expected risk $r_{\exp}(f; u)$ of our classifier on the uniform distribution, the following result upper bounds the DD risk as a function of $\gamma$:

**Theorem 3.2.** *The DD risk of a classifier under the zero-one loss is at most*

$$
r_{dd}(f; \gamma) \leq \min \left\{ \frac{\gamma - \log \left( \frac{1-\alpha}{1-r_{exp}(f; u)} \right)}{\log \left( \frac{\alpha}{1-\alpha} \right) + \log \left( \frac{1}{r_{exp}(f; u)} - 1 \right)}, \; r_{exp}(f; u) + \sqrt{\frac{\gamma}{2}} \right\},
$$

*where $\alpha \in (r_{exp}(f; u), 1)$ may be chosen freely. The DD risk is below 1 for $r_{exp}(f; u) < e^{-\gamma}$.*

We defer the proof to Appendix F. Our experiments confirm that Theorem 3.2 is non-vacuous.

To gain intuition, we set $\alpha = \frac{1}{2}$ and make further simplifications to obtain the following simpler (but less tight) expression:

$$
r_{\mathrm{dd}}(f; \gamma) \leq \min \left\{ \frac{\gamma + \log(2)}{-\log \left( r_{\exp}(f; u) \right)}, \; r_{\exp}(f; u) + \sqrt{\frac{\gamma}{2}} \right\}. \tag{7}
$$

For convenience, we refer to the two arguments to $\min$ in this bound based on their dependency on the expected risk, as "inverse-negative-logarithmic" (first argument) and "additive" (second argument). The additive bound is more informative for smaller entropy gaps $\gamma$. Indeed, the bound reveals that the DD-uniform gap tends to 0 as $\gamma \to 0$. On the other hand, the inverse-negative-logarithmic bound captures more closely the behavior of the DD risk as the expected risk tends to zero, since the function $h(x) = 1/(-\log(x))$ also approaches zero. More generally, our analysis shows that the uniform expected risk should be below $e^{-\gamma}$ to ensure that the DD risk is small, pointing towards a curse of dimensionality unless the test distribution is sufficiently diverse. Specifically, we cannot expect to have a model that is robust to any diverse test distribution shift unless $\gamma = O(1)$.

## 4 DISTRIBUTIONALLY DIVERSE RISK WITHOUT UNIFORM SAMPLES

Although uniform is worst-case optimal, in practice we have to content with samples $Z = \{z_i\}_{i=1}^n$ with $z = (x, f^*(x))$ and $x$ drawn from some arbitrary training distribution with probability density function $p$. Let us denote by $p_Z$ the empirical measure $p_Z = \sum_{i=1}^n 1\{x = x_i\}/n$ of the training set. In the following, we explore ways to mitigate the effect of non-uniformity in the context of finetuning pretrained models and by input-space rebalancing.

### 4.1 APPROACH 0: HOPE THAT $p$ IS CLOSE TO UNIFORM

Before considering any solutions, let us quantify how large the DD risk can be when we train our model on a distribution different from the uniform. We can derive a simple bound on the difference between expected risks of two distributions by the $\ell_1$ distance $\delta(u, p)$ between their densities:

$$
r_{\exp}(f; u) - r_{\exp}(f; p) = \int_x u(x) \, l\left( f(x), f^*(x) \right) dx - \int_x p(x) \, l\left( f(x), f^*(x) \right) dx
$$

$$
= \int_x \left( u(x) - p(x) \right) l\left( f(x), f^*(x) \right) dx \leq \int_x |u(x) - p(x)| dx = \delta(u, p), \tag{8}
$$

where w.l.o.g. we assume that the loss is bounded by 1, such as in the case of the 0-1 loss function. By plugging this result within Theorem 3.2 we find that the DD risk will not change significantly if we train and validate our model on some density $p$ that is very similar to $u$. However, we cannot guarantee anything when the densities differ.

We should also remark that perturbation bounds of the form proposed above might not correlate with empirical observations. Specifically, Ben-David et al. (2006) performed a similar analysis in the context of domain adaptation, showing that $r_{\exp}(f; q) \leq r_{\exp}(f; p) + d_H(p, q) + \lambda$, where $d_H$ is the $H$-divergence between the training $p$ and test density $q$ and $\lambda$ measures the closeness of the respective domains. However, the empirical analysis of Vedantam et al. (2021) "indicates that the theory cannot be used to great effect for predicting generalization in practice".

## 4.2 APPROACH 1: GENTLE FINETUNING

We next focus on finetuning and argue that, independently of how close the training distribution $p$ is to uniform, one may still control the DD risk by controlling the distance between the pretrained model at initialization and the fine-tuned model in the weight space.

Concretely, we adopt a PAC-Bayesian perspective (Alquier et al., 2024) and suppose that the learner uses the training data $Z$ to determine a distribution $\pi_Z : \mathcal{F} \to [0, 1]$ over classifiers $f \in \mathcal{F}$ (equivalently over model weights). We will also assume a prior density $\pi$ that is *unbiased*, meaning that for any $x$ and every $y$, we have $\pi(f(x) = y) = 1/|\mathcal{Y}|$. In Appendix G we prove:

**Theorem 4.1.** *For any unbiased prior $\pi$, the DD risk of a stochastic learner is at most*

$$r_{dd}(\pi_Z; \gamma) := \max_{q \in \mathcal{Q}_\gamma} \mathbb{E}_{f \sim \pi_Z}[r_{exp}(f, q)] \leq \mathbb{E}_{f \sim \pi_Z}[r_{exp}(f, p_Z)] + 2\,\delta(\pi_Z, \pi),$$

*where $l$ is a loss such that $\sum_{y \in \mathcal{Y}} l(y, y') = \sum_{y \in \mathcal{Y}} l(y, y'') \ \forall y', y'' \in \mathcal{Y}$, such as the zero-one loss.*

This result suggests that minimal finetuning on $p_Z$ helps maintain robustness against o.o.d. shifts by not over-fitting the finetuning training set. To make the connection with finetuning more concrete, we remark that the unbiased prior considered may correspond to that induced by a pretrained model. The stochasticity of the prior $\pi$ may stem from the initialization of a readout layer or of low-rank adapters (Hu et al.), mixout (Lee et al.), or may correspond to a Gaussian whose covariance is given by the weight Hessian as in elastic weight consolidation (Kirkpatrick et al., 2017). The posterior $\pi_Z$ can be chosen as the ensemble obtained by repeated (full or partial) finetuning, as is common in the domain generalization literature (Wortsman et al., 2022; Rame et al., 2022; Pagliardini et al.; Rame et al., 2023). Theorem 4.1 states that the DD risk will be close to the empirical risk if finetuning does not change the (posterior over) weights significantly. Intuitively, the effect of a non-uniformly chosen finetuning set to an o.o.d. set can be expected to be small when the ensemble weights remain close to initialization.

Note that our results differ from standard PAC-Bayes generalization bounds (Alquier et al., 2024) as we consider the gap w.r.t. $p_Z$ and the worst distribution in $\mathcal{Q}_\gamma$ rather than the training density $p$. However, both theories correlate distance between prior and posterior with better generalization in the i.i.d. and o.o.d. settings, respectively. Both theories are also subject to a trade-off between the ability to fit the training and test data. PAC-Bayesian arguments are usually applied when training a network from scratch. However, we argue that it can be more relevant to employ them within the context of finetuning. The emergence of foundation models has shown that it is often possible to learn general representations that can act as strong priors on any downstream task. The more powerful the foundation model is, the better the prior, and the more plausible it becomes to fine-tune a model without deviating much from the pretrained weights.

From a practical standpoint, one criticism of Theorem 4.1 is that it is impractical to estimate the $\ell_1$ distance in practice. This may be partially mitigated by relying on the known inequality $\delta(\pi, \pi_Z) \leq \sqrt{2 D_{\mathrm{KL}}(\pi, \pi_Z)}$ to bound the distance in terms of the KL divergence. Further, though the theory discusses stochastic predictors, in practice the benefits of gentle finetuning (i.e., models whose weights have not veered far from initialization) may transfer also to deterministic models. This will be tested empirically by using distance to initialization as an early stopping criterion.

## 4.3 APPROACH 2: TRAINING AND VALIDATION SET REBALANCING

We finally consider the scenario where we use a separate model $w : \mathcal{X} \to \mathbb{R}_+$ trained on a held-out set drawn from $p$ to estimate weights $w(x)$. These weights are now used to rebalance the training and validation set of our classifier leading to the following empirical risk:

$$r_w(f, p_Z) = \frac{1}{n} \sum_{i=1}^{n} w(x_i)\, l(f(x_i), y_i), \tag{9}$$

The choice $w(x) \propto u(x)/p(x)$ is an instance of importance sampling. Importance weights are employed in domain adaptation when the test distribution $q$ was known, whereas we exploit Theorem 3.1 to re-weight towards the uniform. In Appendix I we apply results from importance sampling (Chatterjee & Diaconis, 2018) to characterize the number of *validation* samples needed to

accurately estimate the convergence of $r_w(f, p_Z)$ to $r_{\exp}(f, u)$, where $f$ is chosen independently of $Z$. In our experiments, we will make use of validation set rebalancing for early stopping.

We next account for training set rebalancing by considering instances where $f$ *does* depend on $Z$, whereas $w$ is any general weighting function:

**Theorem 4.2.** *For any Lipschitz continuous loss $l : \mathcal{X} \to [0, 1]$ with Lipschitz constant $\lambda$, weighting function $w : \mathcal{X} \to [0, \beta]$ independent of the training set $Z = (x_i, y_i)_{i=1}^n$, and any density $p$, we have with probability at least $1 - \delta$ over the draw of $Z$:*

$$r_{exp}(f; u) \leq r_w(f; p_Z) + (\beta\lambda\,\mu + \|w\|_L)\,\mathbb{E}_{Z \sim p^n}\left[W_1(p, p_Z)\right] + 2\beta\,\sqrt{\frac{2\ln(1/\delta)}{n}} + \delta(u, \hat{u}),$$

*where the classifier $f : \mathcal{X} \to \mathcal{Y}$ is a function dependent on the training data with Lipschitz constant at most $\mu$, $\delta(u, \hat{u}) = \int_x |u(x) - p(x)w(x)|dx$ is the $\ell_1$ distance between the uniform distribution $u$ and the re-weighted training distribution $\hat{u}(x) = p(x)w(x)$, $W_1(p, p_Z)$ is the 1-Wasserstein distance between $p$ and the empirical measure $p_Z$, and $\|w\|_L$ is the Lipschitz constant of $w$.*

The proof is provided in Appendix H. Similarly to recent generalization arguments (Chuang et al., 2021; Loukas et al., 2024), the proof relies on Kantorovich-Rubenstein duality to capture the effect of the data distribution $p$ through the Wasserstein distance $W_1(p, p_Z)$ between the empirical and expected measures (subsuming analyses that make manifold assumptions). Akin to previous results, the more concentrated $p$ is, the faster the convergence of the empirical measure $p_Z$ will be, implying better i.i.d. generalization (left-most term on the RHS). In addition, a less expressive classifier (quantified by the Lipschitz constant $\mu$) will require fewer samples to generalize.

Where Theorem 4.2 differs from previous arguments (Chuang et al., 2021; Loukas et al., 2024) is that it bounds the rebalancing gap $r_{\exp}(f; u) - r_w(f; p_Z)$ (relevant to DD risk as per Theorem 3.2) rather than the typical generalization gap $r_{\exp}(f; p) - r(f; p_Z)$. By selecting $w(x) \propto 1/p(x)$ we can control the $\ell_1$ distance term $\delta(u, \hat{u})$, but this may be at the expense of worse i.i.d. generalization if the maximum weight value $\beta$ and the Lipschitz constant $\|w\|_L$ is increased as a result.

From a practical perspective, the theorem suggests two weighting functions that balance in- and out-of-distribution: enforced upper bound $w(x) = \min\{1/p(x), \beta\}$ and enforced smoothness $w(x) = p(x)^\tau$ with $\tau \leq 1$. The two choices discount the effect of $\|w\|_L$ and $\beta$, respectively, by assuming that they do not correspond to the dominant factor in the bound. Note also that, since $u(x)$ is a constant, optimizing the model parameters with gradient-based methods will result to the same solution independently of whether one includes $u(x)$ on the numerator of $w(x)$ or not. Clipping is especially important when there are outliers due to noise and thus $1/p(x)$ is very large.

## 5 EXPERIMENTS

The following experiments to validate our theory empirically, examine the effectiveness of the approaches considered in improving o.o.d. generalization, and identify pitfalls. In summary they demonstrate that, when the training density can be fit aptly, rebalancing consistently improves performance in scenarios with significant covariate shift.

### 5.1 THEORY VALIDATION IN A CONTROLLED EXPERIMENTAL SETUP

We start by testing our theoretical predictions on a mixture of Gaussians classification task, where each mode is assigned a random class label and the goal of the classifier is to classify each point based on whether its likelihood ratio is above or below 1. We train a multi-layer perceptron on either a uniform or non-uniform training distribution, with a training set size $n$ ranging from 100 to 10,000 examples. To obtain confidence intervals, in the following we sample 35 tasks and repeat the analysis for each one. The controlled setting allows for precise evaluation of entropy, accurate approximation of the worst-case distribution, and exploration of the limits of data. Further information about the experimental setup can be found in Appendix C.2.

We first examine how the DD risk evolves as a function of the training set size when the model is trained on a uniform distribution. We approximate the risk of the worst-case distribution $q^* = \arg\max_{q \in \mathcal{Q}_\gamma} r_{\exp}(f; q)$ using a greedy adversarial construction (see Appendix C.2). The DD risk

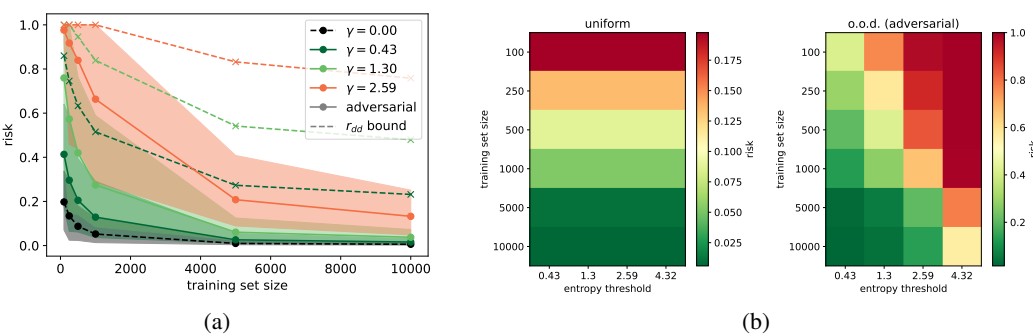

(a)                                                (b)

Figure 1: Influence of training set size and entropy gap on DD risk $r_{dd}(f; \gamma)$ on the mixture of Gaussians task. Here the DD risk is greedily approximated by constructing adversarial test distributions that satisfy the desired entropy bound. The number of training data required to achieve a low DD risk increases sharply with the entropy gap $\gamma$ between the uniform and the test distribution, interpolating between the uniform expected risk and the worst-case risk. The adversarial test distribution risk is always below our $r_{dd}$ bound from Theorem 3.2.

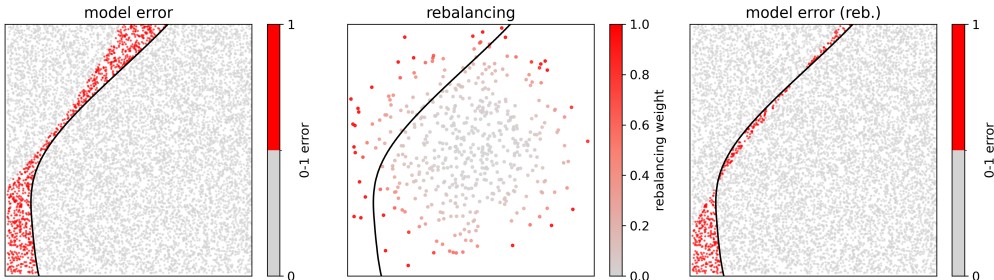

Figure 2: Effect of rebalancing on model error. Left: In red, we depict the area over which the model predicts the wrong label when trained without rebalancing. The black line denoted the ground-truth decision boundary. Middle: The plot shows the training set (sampled from a Gaussian distribution) and the importance weights used for rebalancing. These focus the model's attention to more sparsely sampled regions. Right: When trained with rebalancing, the model approximates more closely the ground-truth decision boundary.

is then approximated by the risk of the classifier on the adversarial set. The plotted shaded regions indicate the 5-th and 95-th risk percentiles across repeats. As shown in Figure 1, the empirical (and approximate) DD risk decreases more rapidly for smaller entropy gaps as the number of samples increases. The theoretical bound is non-vacuous and tracks the performance against the o.o.d. test distribution. The remaining gap between theory and practice can be partially explained by the fact that we employed a greedy construction to construct the test distribution that provides a $1 - 1/e$ approximation of the true optimum giving rise to the DD risk.

Next, we examine the impact of non-uniformity in the training distribution. We select a Gaussian training distribution centered at the center of the domain with an increasing standard deviation, truncated to the unit square. We fix $n = 500$ and vary $\sigma$. As expected, Figure 3 shows that the DD risk decreases as $\sigma$ increases, indicating that broader coverage of the domain improves generalization. We also investigate how well rebalancing can mitigate the effects of non-uniformity. Figure 3 shows that by controlling the re-weighting strategy it is possible to improve o.o.d. generalization, thereby partially overcoming the challenges posed by non-uniform training sets. We describe how we implement reweighting in practice in Appendix C.1.

To gain intuition, we also plot the model output on a specific task instance in Figure 3 with the true decision boundary shown in black. The left- and right-most sub-figures show the miss-classifier points in red, respectively without and with rebalancing. The training set is Gaussian-distributed and can be seen in the middle sub-figure. This model exhibits poor test error for distribution that assign

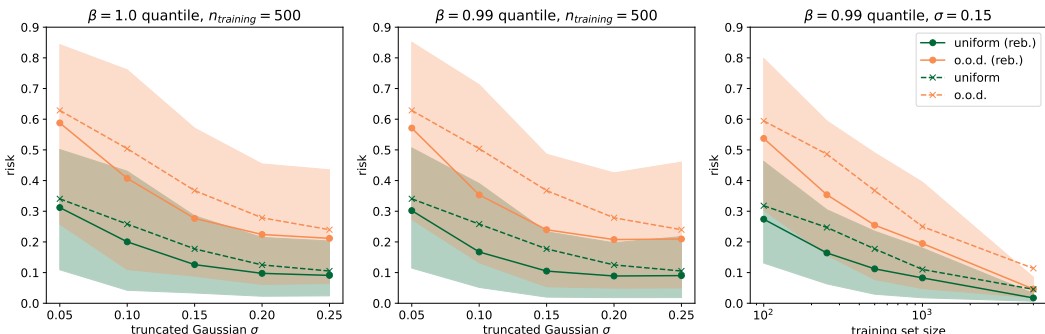

Figure 3: The achieved DD risk is smaller for models trained on more uniform training data. The training data is drawn from a truncated Gaussian distribution with increasing standard deviation, such that the sampling becomes gradually more uniform over our sample space. As theorized, the DD risk decays for larger $\sigma$, following the trend of the uniform expected risk. rebalancing reduces uniform expected and DD risk risk (here for $\gamma = 0.99$). We use a masked auto-regressive flow $\hat{p}$ to fit the density $p$ of the training data and set $w(x) \propto \min(1/\hat{p}(x_i)^{\tau}, \beta)$, with $\tau = 1$ controlling the smoothness of the weights and $\beta$ set based on a quantile of the training likelihood capping the effect of outliers. Naturally, increasing dataset size reduces DD risk. However, rebalancing remains equally beneficial across all training set sizes tested, showing that increase in data size does not remove the need for uniformity.

Table 1: *iWildCam*. Macro F1 and average classification accuracy (higher is better). o.o.d. results are on images from wildlife cameras not present in the training set, while i.d. results are from the cameras in the training set taken on different days. Parentheses show standard deviation across 3 replicates. We modified C-Mixup* (Yao et al., 2022) for categorical labels (see Appendix C.4).

| | Validation (i.d.) | | Validation (o.o.d.) | | Test (i.d.) | | Test (o.o.d.) | |
|---|---|---|---|---|---|---|---|---|
| Algorithm | Macro F1 | Avg acc | Macro F1 | Avg acc | Macro F1 | Avg acc | Macro F1 | Avg acc |
| ERM | 48.8 (2.5) | 82.5 (0.8) | 37.4 (1.7) | 62.7 (2.4) | 47.0 (1.4) | 75.7 (0.3) | 31.0 (1.3) | 71.6 (2.5) |
| CORAL | 46.7 (2.8) | 81.8 (0.4) | 37.0 (1.2) | 60.3 (2.8) | 43.5 (3.5) | 73.7 (0.4) | 32.8 (0.1) | 73.3 (4.3) |
| IRM | 24.4 (8.4) | 66.9 (9.4) | 20.2 (7.6) | 47.2 (9.8) | 22.4 (7.7) | 59.9 (8.1) | 15.1 (4.9) | 59.8 (3.7) |
| Group DRO | 42.3 (2.1) | 79.3 (3.9) | 26.3 (0.2) | 60.0 (0.7) | 37.5 (1.7) | 71.6 (2.7) | 23.9 (2.1) | 72.7 (2.0) |
| C-Mixup* | 44.1 (0.8) | 80.5 (0.7) | 33.1 (0.6) | 57.2 (2.6) | 43.1 (0.9) | 71.9 (0.5) | 26.8 (1.4) | 70.2 (2.5) |
| Label reweighted | 42.5 (0.5) | 77.5 (1.6) | 30.9 (0.3) | 57.8 (2.8) | 42.2 (1.4) | 70.8 (1.5) | 26.2 (1.4) | 68.8 (1.6) |
| Rebalancing | 49.1 (1.5) | 82.8 (1.7) | 38.8 (0.7) | 62.7 (0.6) | 48.1 (3.1) | 76.1 (1.0) | 31.5 (1.8) | 71.6 (1.2) |
| Rebalancing (PCA-256) | 51.4 (1.8) | 83.9 (1.3) | 39.7 (0.6) | 65.4 (0.5) | 47.0 (3.1) | 76.7 (1.2) | 33.5 (1.0) | 75.3 (0.6) |
| Rebalancing (label cond.) | 54.0 (1.5) | 84.5 (1.4) | 39.5 (0.5) | 66.7 (1.5) | 50.1 (1.8) | 77.5 (1.1) | 34.9 (1.1) | 77.4 (0.8) |
| Rebalancing (PCA, label cond.) | 53.9 (0.7) | 84.2 (0.4) | 40.2 (0.8) | 66.5 (1.5) | 49.8 (1.4) | 77.0 (0.3) | 35.5 (0.8) | 75.3 (2.8) |

higher probability of the domain boundaries. rebalancing mitigates this effect leading to a tighter approximation of the true function across the entire domain and thus improved o.o.d. test error.

## 5.2 MITIGATING COVARIATE SHIFT IN PRACTICE

We proceed to evaluate generalization on various classification tasks involving o.o.d. shifts from popular benchmarks (Koh et al., 2021; Gulrajani & Lopez-Paz, 2021). We select tasks focusing primarily on those that involve covariate shift rather than concept, domain, or label drift. We compare to vanilla empirical risk minimization (ERM) and the baselines reported by the original studies; we do not claim state-of-the-art performance. To examine the effect of rebalancing, we require a density estimator. After experimentation in the mixture of Gaussians task, we settled in favor of masked auto-regressive flow (MAF) (Papamakarios et al., 2017a) fit on the embeddings of the pretrained model that is then fine-tuned to solve the task at hand. Further details can be found in Appendix C.1.

Tables 1, 2, and 3 present the results on the iWildCam (Beery et al., 2021), PovertyMap (Koh et al., 2021), and ColorMNIST (Arjovsky et al., 2019) tasks, respectively. Our theory motivates rebalancing as a strategy for improving worst-group performance and, indeed, we observe higher gains for the worst-group in ColorMNIST and PovertyMap (iWildCam has no such split).

Table 2: *PovertyMap.* Pearson correlation (higher is better) on in-distribution and out-of-distribution (unseen countries) held-out sets, incl. rural subpopulations. All results are averaged over 5 different o.o.d. country folds, with standard deviations across different folds in parentheses.

| Algorithm | Validation (i.d.) | | Validation (o.o.d.) | | Test (i.d.) | | Test (o.o.d.) | |
|---|---|---|---|---|---|---|---|---|
| | Overall | Worst | Overall | Worst | Overall | Worst | Overall | Worst |
| ERM | 0.82 (0.02) | 0.58 (0.07) | 0.80 (0.04) | 0.51 (0.06) | 0.82 (0.03) | 0.57 (0.07) | 0.78 (0.04) | 0.45 (0.06) |
| CORAL | 0.82 (0.00) | 0.59 (0.04) | 0.80 (0.04) | 0.52 (0.06) | 0.83 (0.01) | 0.59 (0.03) | 0.78 (0.05) | 0.44 (0.06) |
| IRM | 0.82 (0.02) | 0.57 (0.06) | 0.81 (0.03) | 0.53 (0.05) | 0.82 (0.02) | 0.57 (0.08) | 0.77 (0.05) | 0.43 (0.07) |
| Group DRO | 0.78 (0.03) | 0.49 (0.08) | 0.78 (0.05) | 0.46 (0.04) | 0.80 (0.03) | 0.54 (0.11) | 0.75 (0.07) | 0.39 (0.06) |
| C-Mixup | 0.84 (0.01) | 0.64 (0.05) | 0.81 (0.04) | 0.55 (0.06) | 0.85 (0.01) | 0.64 (0.05) | 0.80 (0.04) | 0.51 (0.08) |
| Rebalancing | 0.83 (0.01) | 0.62 (0.02) | 0.80 (0.03) | 0.53 (0.04) | 0.84 (0.02) | 0.63 (0.04) | 0.75 (0.07) | 0.44 (0.06) |
| Rebalancing (UMAP-64) | 0.85 (0.01) | 0.66 (0.03) | 0.80 (0.03) | 0.53 (0.04) | 0.85 (0.01) | 0.65 (0.04) | 0.78 (0.04) | 0.47 (0.10) |

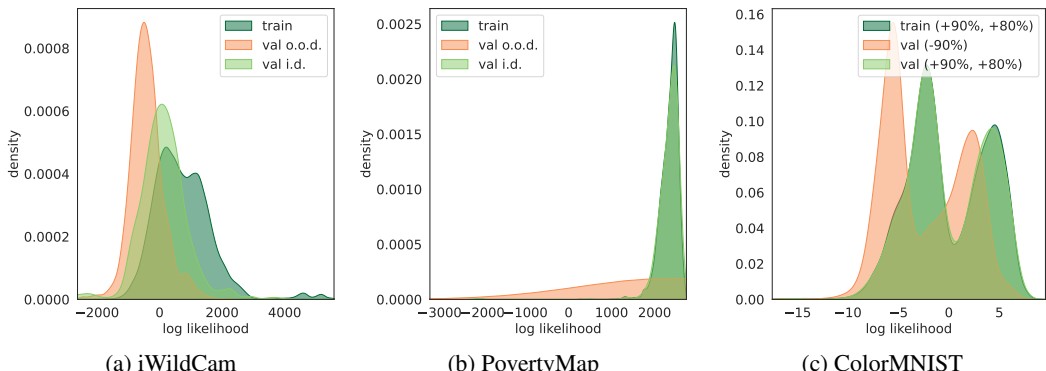

(a) iWildCam      (b) PovertyMap      (c) ColorMNIST

Figure 4: Log-likelihoods used for training set rebalancing as well as i.d. and o.o.d. set log-likelihood distributions. iWildCam and ColorMNIST feature covariate shift, as the density support is largely the same across all sets. The o.o.d. PovertyMap set contains a notable domain shift.

Interestingly, in both PovertyMap and iWildCam, rebalancing yields performance improvements also in the in-distribution (i.d.) sets. Closer inspection reveals that the i.d. validation and test sets for iWildCam were not sampled i.i.d.. This can be also seen in Figure 4 which depicts the estimated log-likelihoods distributions for the training, i.d., and o.o.d. sets. These plots confirm a gradual increase in covariate shift from i.d. to o.o.d. validation sets. On the other hand, inspection of the PovertyMap o.o.d. set likelihoods reveals a noticeable domain shift for a large fraction of the set, which explains why rebalancing is less effective in this instance.

While our base approach often results in improvements, we found that better results could be achieved by introducing a dimensionality reduction step prior to density estimation (UMAP McInnes et al. (2018) or PCA) or by fitting a label-conditioned density on the training set. Both are described in Appendix C.1. The best configuration was selected through ablation over relevant hyperparameters. However, the larger number of moving parts reveals the brittleness of our MAF density estimator, which influences the gains achieved. This issue is further discussed in Section 5.3.

We further explore the effect of gentle finetuning by modifying the model selection process in ColorMNIST. We focus on the -90% group which features the largest covariate shift. In DomainBed it is standard practice to use a held-out training subset for early stopping. As shown in Table 3, removing early stopping but using the held out set for hyperparameter selection slightly improves the performance of ERM, indicating a substantial mismatch between the training and worst-group distributions. Performance improves further when we use the weight distance to initialization (WDL2) instead of validation error to select model configurations, as motivated by our gentle finetuning analysis in Section 4.2, also with no early stopping. Further gains are observed when we add rebalancing of the training set while also using the WDL2 model selection. The best result was obtained by combining the above with dimensionality reduction prior to fitting the density estimator. WDL2 was used only in ColorMNIST as in WILDS benchmarks it is a convention to use an appropriate o.o.d. validation set for model selection. When such a set is available, it is the preferred choice. See Appendix C.3 for further investigation of the impact rebalancing has on different hyperparameters.

Table 3: ColoredMNIST. Binary classification accuracy (higher is better). Our methods bring benefits w.r.t. model performance on the group (-90%) that entails the largest covariate shift. Since model selection strategies are crucial for this task, in addition to the official implementation that uses validation-based early stopping (first part of the table), we also test the effect of the following model selection strategies: WDL2 entails using the weight distance to initialization for model selection, motivated by our gentle finetuning argument.

| Algorithm | +90% | +80% | -90% | Avg |
|---|---|---|---|---|
| ERM | $71.7 \pm 0.1$ | $72.9 \pm 0.2$ | $10.0 \pm 0.1$ | 51.5 |
| IRM | $72.5 \pm 0.1$ | $73.3 \pm 0.5$ | $10.2 \pm 0.3$ | 52.0 |
| GroupDRO | $73.1 \pm 0.3$ | $73.2 \pm 0.2$ | $10.0 \pm 0.2$ | 52.1 |
| Mixup | $72.7 \pm 0.4$ | $73.4 \pm 0.1$ | $10.1 \pm 0.1$ | 52.1 |
| MLDG | $71.5 \pm 0.2$ | $73.1 \pm 0.2$ | $9.8 \pm 0.1$ | 51.5 |
| CORAL | $71.6 \pm 0.3$ | $73.1 \pm 0.1$ | $9.9 \pm 0.1$ | 51.5 |
| MMD | $71.4 \pm 0.3$ | $73.1 \pm 0.2$ | $9.9 \pm 0.3$ | 51.5 |
| DANN | $71.4 \pm 0.9$ | $73.1 \pm 0.1$ | $10.0 \pm 0.0$ | 51.5 |
| CDANN | $72.0 \pm 0.2$ | $73.0 \pm 0.2$ | $10.2 \pm 0.1$ | 51.7 |
| MTL | $70.9 \pm 0.2$ | $72.8 \pm 0.3$ | $10.5 \pm 0.1$ | 51.4 |
| SagNet | $71.8 \pm 0.2$ | $73.0 \pm 0.2$ | $10.3 \pm 0.0$ | 51.7 |
| ARM | $82.0 \pm 0.5$ | $76.5 \pm 0.3$ | $10.2 \pm 0.0$ | 56.2 |
| VREx | $72.4 \pm 0.3$ | $72.9 \pm 0.4$ | $10.2 \pm 0.0$ | 51.8 |
| RSC | $71.9 \pm 0.3$ | $73.1 \pm 0.2$ | $10.0 \pm 0.2$ | 51.7 |
| ERM (no early stopping) | $71.1 \pm 0.4$ | $72.8 \pm 0.2$ | $10.2 \pm 0.2$ | 51.4 |
| ERM (WDL2) | $66.9 \pm 0.8$ | $71.4 \pm 0.8$ | $11.0 \pm 0.5$ | 49.8 |
| Rebalancing (no early stopping) | $71.6 \pm 0.4$ | $72.6 \pm 0.4$ | $10.1 \pm 0.2$ | 51.4 |
| Rebalancing (UMAP-8, label cond., no early stop.) | $70.4 \pm 0.3$ | $73.6 \pm 0.3$ | $10.8 \pm 0.3$ | 51.6 |
| Rebalancing (WDL2) | $70.7 \pm 0.4$ | $70.9 \pm 2.0$ | $12.0 \pm 1.0$ | 51.2 |
| Rebalancing (UMAP-8, label cond., WDL2) | $69.5 \pm 1.0$ | $72.7 \pm 0.7$ | $37.0 \pm 10.7$ | 59.7 |

## 5.3 Pitfalls

A key prerequisite for rebalancing to work is that we can successfully fit a density over the training set to derive importance weights. Our theory suggests that rendering the training set more uniform can lead to models that are more robust to high entropy test distributions.

We already saw in the experiments above that introducing carefully tuned dimensionality reduction or label conditioning when fitting the density could yield significant benefits. Unfortunately, we also encountered datasets for which the density fit was so poor that the aforementioned modifications did not suffice to improve performance. In Appendix C.5 we take a deeper dive into these failures, showing how the failure of the density estimator impacts test performance.

Finally, we re-iterate that this work focuses on test distributions supported in the same domain as the training distribution. Many tasks in the popular DomainBed and WILDS benchmarks contain significant domain and label shifts which require a different approach.

## 6 Conclusion

As machine learning progresses toward larger-scale datasets and the development of foundation models, it becomes increasingly important to move beyond traditional i.i.d. guarantees and to consider worst-case scenarios in o.o.d. generalization. In this spirit, our work introduces a novel perspective that prioritizes minimizing worst-case error across diverse distributions. We have shown that training on uniformly distributed data offers robust guarantees, making it a powerful strategy as models scale in complexity and scope. Even when uniformity cannot be achieved, we find that rebalancing strategies can provide practical avenues to enhance model resilience.

Further empirical work focusing on obtaining more robust density estimates as well as investigation of gentle finetuning in the context of ensembles and foundation model training could bring additional benefits. Other potential avenues to mitigate training data non-uniformity could include adding noise to the input data (Bishop, 1995) and mixup (Zhang, 2017). We also note that the observations we have made about how our theory aligns with previous evidence in the literature on the training of foundation models do not establish a causal relation between our theory and reality. Further work will be needed to rigorously establish these links.

Overall, the shift in focus from average-case to worst-case generalization presents a compelling framework for the next generation of machine learning models, particularly as they are trained on increasing larger datasets and deployed in increasingly unpredictable environments.

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

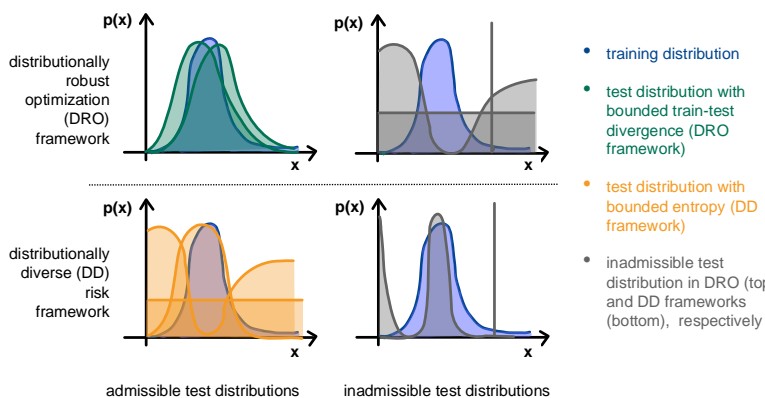

Figure 5: Conceptual illustration of the differences between the distributionally robust optimization (DRO) and distributionally diverse risk (DD) frameworks, respectively shown in the top and bottom rows. We here consider the example of a 1-dimensional density. The training distribution is given in blue, whereas in green and orange we depict example admissible test distributions according to DRO and DD, respectively: DRO assumes that the test distribution will be close to the training one (small train-test divergence). In the DD framework, the admissible test distributions have high entropy but may have arbitrarily large train-test divergence. In gray, we depict examples of non-admissible distributions for DRO (top) and DD (bottom). We provide examples of admissible distributions for the DD framework that are inadmissible in DRO and vice-versa.

## A  RELATED WORK

Out-of-distribution generalization is a central challenge in machine learning, where models trained on a specific data distribution are required to perform well on unseen distributions that may differ significantly. For an in-depth survey of the flurry of work on o.o.d. generalization, we refer to the following surveys (Liu et al., 2023; Zhou et al., 2022; Wang et al., 2022). In the following, we discuss the various approaches that have been developed to this challenge, emphasizing their similarities and differences with this work.

**Domain generalization.** Domain generalization and invariant risk minimization aim to enhance a model's ability to generalize across different environments or domains. These methods define a set of environments from which data is gathered and seek to ensure that the model's outputs are invariant (i.e., indistinguishable) with respect to the environment from which the data originated (Bengio et al., 2013; Peters et al., 2016; Arjovsky et al., 2019; Rosenfeld et al., 2020; Koyama & Yamaguchi, 2020). The ultimate goal is to capture features that are discriminative across domains while ignoring spurious correlations. Similarly to these methods, our work acknowledges potential shifts between training and test distributions and aims to promote robust generalization. Unlike domain generalization approaches, we do not assume that the data comes from multiple pre-defined environments. Recent advances, particularly those that combine finetuning of pretrained models with ensembling strategies, are relevant to our approach as they have shown promise in enhancing generalization across varied domains (Gulrajani & Lopez-Paz, 2021). Ensemble-based methods such as model soups (Wortsman et al., 2022), DiWA (Rame et al., 2022), agree-to-disagree (Pagliardini et al.), and model Ratatouille (Rame et al., 2023) have further extended these concepts, aligning closely with our objectives.

**Robustness to uncertainty and perturbation.** Robust optimization (RO) is a well-established field in optimization that deals with uncertainty in model parameters or data (Ben-Tal et al., 2009). In the context of machine learning, RO has been applied to scenarios where the test distribution is assumed to be within a set of known plausible distributions, with the goal of minimizing the worst-case loss over this set (Caramanis et al., 2011; Singla et al., 2020; Zhang et al., 2022). Distributionally robust optimization (DRO) (Rahimian & Mehrotra, 2019; Duchi & Namkoong, 2019; 2021) extends this concept by considering a set of distributions over the unknown data, usually inferred through a prior

on the data, such as distributions that are close to the training distribution by some distance measure such as the Wasserstein distance (Kuhn et al., 2019). Our approach shares similarities with DRO in that the DD risk in that we both consider the worst case behavior over a set of distributions and that, at the limit, both DRO and DD converge to the worst case risk over the domain. However, our work diverges from traditional RO and DRO frameworks in significant ways. RO typically focuses on the optimization aspect, often assuming a convex cost function and providing theoretical justification for various regularizers such as Tikhonov and Lasso (Caramanis et al., 2011). These approaches are particularly concerned with uncertainty within a bounded region near the training data, such as noisy or partial inputs and labels (Singla et al., 2020; Zhang et al., 2022). In contrast, our focus is on scenarios where the training data is neither noisy nor partial but where the test distribution can change almost arbitrarily within the support of the training distribution, provided that its entropy is not too small. Critically, as illustrated in Figure 5, the assumptions posed by DRO and our framework about the relation of the train and test distribution are different: we do not consider the worst-case bounded distribution shift (as is done in DRO) but the worst case risk under *any* distribution of sufficient entropy. As such, the training distribution holds no special role in our framework, whereas in practice DRO defines the set of potential test distributions as those some distance away from it. This is a consequential conceptual and practical difference: Conceptually, whereas in DRO one needs to think about how similar is the test distribution to the training distribution, we here consider arbitrary distributions on the test domain and pose a constraint on entropy. Practically, the two approaches lead to very different solutions to the problem of distribution shift. Within the deep learning community, significant efforts have also been made to train models that are robust to small input perturbations, referred to as adversarial examples (Goodfellow et al., 2015; Madry et al., 2018; Alayrac et al., 2019; Bai et al., 2021). While these methods have influenced our understanding of robustness, our work is distinct in that we address broader shifts in the test distribution rather than specific adversarial perturbations.

**Domain adaptation and covariate shift.** Domain adaptation is a sub-area of transfer learning (Pan & Yang, 2009) focused on transferring knowledge from a source domain to a target domain where the data distribution differs (Farahani et al., 2021). This is crucial when a model trained on one domain is expected to perform well on a new domain. Domain adaptation falls into three types of domain shifts: covariate, concept, and label shift. Our work is most closely related to the covariate shift scenario, in which the distribution of input features changes between the training and test phases. Similarly to some domain adaptation methods that address covariate shift, we also consider re-weighting strategies to account for differences between training and test distributions (Shimodaira, 2000; Huang et al., 2006) (other approaches consider ensemble disagreement (Jiang et al.; Kirsch & Gal), model confidence (Garg et al., 2022), and neighborhood invariance (Ng et al.) on the unlabeled data). A key difference is that we do not assume that we have access to unlabeled target domain data but instead consider re-weighing to uniformly rebalance the training set. This choice motivated the proven optimality of the uniform distribution when considering the worst-case scenario across all possible distributions that are constrained by a given entropy threshold. This broader DD framework enables robust generalization across diverse scenarios, without relying on prior knowledge of specific target distributions or adaptations tailored to known shifts.

**Active learning.** Active Learning (AL) aims to efficiently train models by selecting the most informative data points for labeling, thereby reducing the amount of labeled data needed to achieve high performance (Cohn et al., 1996; Settles, 2009). Within AL, entropy maximization is a common practice, where the focus is on maximizing the entropy of the predictive distribution $p(y|x)$ to identify the most uncertain and thus informative data points. Other acquisition functions include Variation Ratios (which select data points based on the disagreement among multiple model predictions), mean standard deviation (Kendall et al., 2017) provide alternative ways to measure uncertainty. Additionally, methods like mutual information between predictions and model posterior have been used to target data points that most influence the model's posterior distribution (Houlsby et al., 2011). The use of density estimation to bias the sampling towards low-density areas was also considered in (Zhu et al., 2008). Therein, they define the density$\times$entropy measure which combines $H(y|x)$ and $p(x)$. For a recent overview of these sampling strategies in computer vision, refer to Gal et al. (Gal et al., 2017). Despite the overlap in techniques, our work focuses on o.o.d. generalization rather than iterative sample acquisition: while AL aims to improve model performance by selecting specific data points during training, our study seeks to optimize generalization across all possible distributions within a domain, offering a different perspective on managing uncertainty.

## B  JUSTIFICATION OF THE DD RISK FORMULATION

The DD risk, that we defined as follows:

$$r_{\text{dd}}(f; \gamma) = \max_{q \in \mathcal{Q}_\gamma} \mathbb{E}_{x \sim q}\Big[ l\big(f(x), f^*(x)\big)\Big],$$

measures the behavior of a classifier across any test distribution of sufficiently high-entropy within a domain. In the following, we justify this definition from a mathematical, intuitive, and empirical perspective.

**Mathematical justification.** Our definition of DD risk follows from two desiderata about how to model the generalization of a predictive model under covariate shift:

1. *We know very little about the relation between the training and test distributions.*
2. *We wish to avoid judging the behavior of a classifier based on pathological examples, thus the test distribution should not assign high likelihood to any small set of examples.*

Desideratum 1 is posed because the distance between training and test distribution (also referred to as *train-test discrepancy*) is often large in practice. This is supported by the literature where, train-test discrepancy has been empirically found to correlate poorly with generalization (Vedantam et al., 2021) and concurrent theoretical work also advocates against it (Bhattacharjee et al., 2024). On the other hand, desideratum 2 addresses the reason why worst-case analysis is non-meaningful in learning: if no such constraint is placed, an adversary may always construct a pathological test distribution that concentrates all its mass on a small set of inputs where the predictive model is wrong. We directly avoid this situation by asserting that no such small high likelihood set can exist.

Asserting that the test distribution entropy is high satisfies both these desiderata without imposing any further assumptions. Lower bounding the test entropy ensures that the spread of a distribution is large and does not constrain the shape of the test distribution based on that of the training one given a fixed domain.

**Intuitive explanation.** We argue that, a generally performant model is one that generally performs well on many test instances that the world throws at it—even if it might fail in specific instances. In other words, though we cannot hope that our model will generalize to any arbitrary sharp test distribution, we can reasonably expect that it will performs well on test distributions that are more spread out and thus more typical of the data domain. Our assumption of high test entropy exactly corresponds to the minimal and intuitive assumption that the classifier will be evaluated on a diverse set of possible inputs, rather than on few pathological examples.

**Empirical justification.** Finally, to emphasize the practical applicability of our assumption in real-world applications, we remark that, as mentioned in the introduction, test entropy has already been identified in the literature (Vedantam et al., 2021) as an intuitive and empirically predictive measure for o.o.d. generalization.

## C  ADDITIONAL EXPERIMENTAL DETAILS

### C.1  REBALANCING IN PRACTICE

The aim of rebalancing is to fit a density function to the training examples $p(x)$ such that, after rebalancing, the training examples with weights $w(x) \propto 1/p(x)^\tau$ resemble a uniform distribution. While temperature $\tau$ can be tuned as a hyperparameter, we kept $\tau = 1.0$ for simplicity. As per Theorem 4.2, we cap the weights $w(x) \propto \min(1/\hat{p}(x_i)^\tau, \beta)$. In practice this is used as a way to regularize the distribution and avoid oversampling outliers. From synthetic experiments we found $\beta = 0.99$ quantile to be a robust choice and used it in all further real-world experiments.

We use a masked autoregresive flow (MAF) (Papamakarios et al., 2017b) to fit a density to the training set embeddings as we found it performed better than alternatives in preliminary experiments. In all setups we use a standard Gaussian as a base density. For real-world experiments we used MAF with 10 autoregressive layers and a 2 layer MLP with a hidden dimension of 256 for each of them. For synthetic experiments we used MAF with 5 autoregresive layers and a 2 layer MLP with a hidden dimension of 64 for each of them. In all cases, when training MAF we hold out 10% random subset of the training data for early stopping, with a patience of 10 epochs. The checkpoint with the best

held-out set likelihood is used. MAF is trained with learning rate of $3e - 4$ and the Adam (Kingma, 2014) optimizer.

When using label conditioning, we use the flow to estimate conditional density $p(x) = p_{\text{MAF}}(x|y)p(y)$, where $p(y)$ is computed from label frequency in the training set. Further, in this case we ensure that final weights upsample minority label samples by scaling weights to inverse label frequency $w' = w/p(y)$.

To achieve good quality rebalancing we need the sample weights to capture the data landscape aspects relevant to the given problem. As we focus on generalization tasks where it is standard to use pre-trained models (Koh et al., 2021) as a starting point, we aim to fit a density on the embeddings produced by the pre-trained backbone model. For real world experiments we use the hyperparameters and experimental setup proposed for ERM in the respective original papers (Koh et al., 2021; Gulrajani & Lopez-Paz, 2021).

All WILDS (Koh et al., 2021) datasets considered, except PovertyMap, use a pre-trained model that is finetuned with a new prediction head. We use that pre-trained model as a featurizer to produce the training set embeddings. For PovertyMap, the randomly initialized featurizer is used, to keep in line with the original experimental setup, even though using a model pre-trained on ImageNet marginally improves the results. In all cases the featurizer models used already produce one embedding vector per training sample, except CodeGPT (Lu et al., 2021) used for Py150, for which we apply mean pooling over the sequence to produce a single embedding vector to use for density estimation.

For ColorMNIST traditionally no pre-trained backbone is used (Gulrajani & Lopez-Paz, 2021). Thus, we use a ResNet-50 model pre-trained on ImageNet to build the embeddings for density estimation, but otherwise train the standard model architecture, with exactly the same experimental setup as proposed by Gulrajani & Lopez-Paz (2021).

To combat the curse of dimentionality and to potentially fit smoother densities we also explore dimensionality reduction. In all real-world datasets we perform a small grid search over transforming embeddings using UMAP (McInnes et al., 2018) to 8 or 64 dimensions or using PCA to 256 dimensions before fitting the flow model. We use the validation loss to select the dimensionality reduction technique. We selected on these grid-search hyperparameters from preliminary experiments where we found that at lower target dimensions UMAP performs much better than PCA in our setup, while for larger target dimensions PCA is at least as good. A more exhaustive hyperparameter search was avoided to save computational resources. To facilitate the flow training, we normalize the embeddings. We grid search over two options: normalizing each embedding dimension to unit variance and zero mean or normalizing all embeddings by the maximum vector length. To get a better behaved probability distribution, we scale log-likelihoods produced by MAF by the dimension size, to get likelihoods in bits per dimension, before applying a softmax over the whole training set, to get proper sample weights.

In the synthetic experiments, we directly fit the flow to the data without employing any dimensionality reduction.

## C.2 MIXTURE OF GAUSSIANS

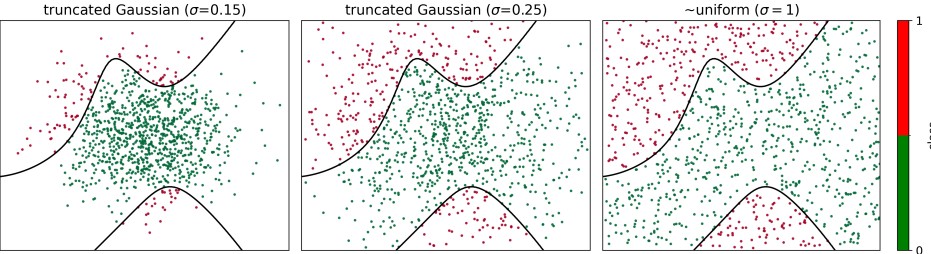

Figure 6: Different sampling strategies for our synthetic dataset. Points are sampled either uniformly or using a truncated Gaussian, with varying standard deviations. To label the samples, we use four randomly placed univariate Gaussian centroids and for each point $x$ assign the label $y$ (red or green) of the Gaussian with the highest likelihood. Resulting decision boundary is in black.

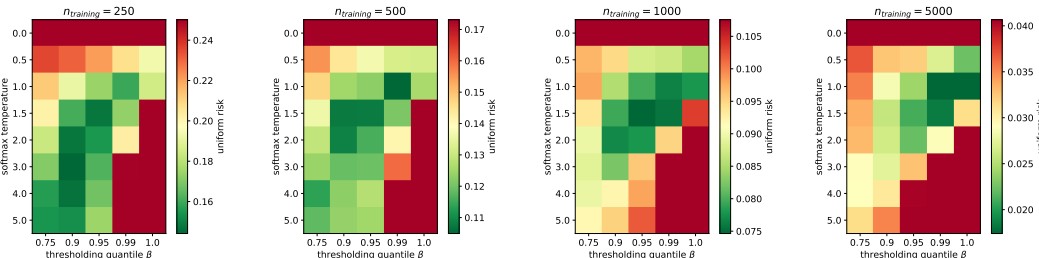

Figure 7: Effect of different threshold $\beta$ and temperature $\tau$ values when using sample rebalancing for training datasets with varying levels of uniformity (truncated gaussian $\sigma$. For visualization, error values are clipped from above to the value achieved without rebalancing.

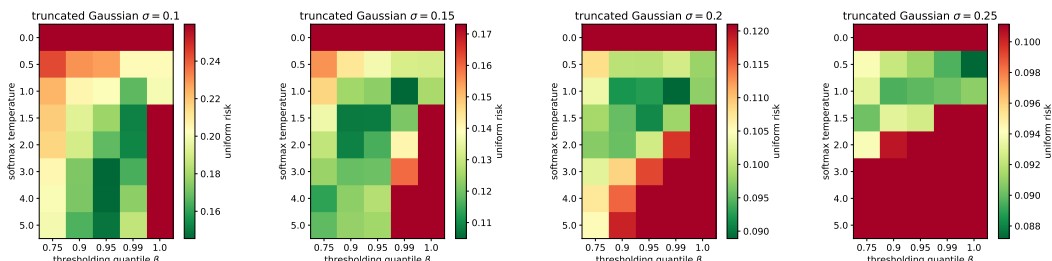

Figure 8: Effect of different threshold $\beta$ and temperature $\tau$ values on expected when using sample rebalancing across training sets with different non-uniformity (captured by the standard deviation $\sigma$). For visualization, error values are clipped from above to the value achieved without rebalancing.

We consider four isotropic Gaussians, where the means are uniformly selected within the unit square $[0, 1]^2$. Two of these Gaussians represent the positive class, and the other half represent the negative class. A point in $[0, 1]^2$ is labeled as positive if the likelihood ratio of positive to negative mixtures exceeds one. We train a multi-layer perceptron on either a uniform or non-uniform training distribution, with a training set size $n$ ranging from 100 to 10,000 examples. To obtain confidence intervals, in the following we sample 35 tasks and repeat the analysis for each one. Illustrative examples of the task are provided in Figure 6.

A key challenge we encountered was the reliable estimation of entropy, as the entropy estimators we tried were biased, a common issue in entropy estimation. To address this, we partition the space into $100{\times}100$ bins $\mathcal{B}$, sample a held-out set of $m = 10,000$ examples, and estimate the discrete entropy using the formula $H(p) \approx \sum_{b \in \mathcal{B}} \hat{p}_b(x) \log(1/\hat{p}_b(x))$, where $\hat{p}_b(x) = \sum_x \mathbb{1}\{x \in b\}/m$.

We select the test distribution using a greedy adversarial construction: first, we sample 10 000 points uniformly, then starting with all test points that the model mislabels, an adversarial set is iteratively expanded by adding the point that maximizes entropy at each step. Whereas set selection for entropy maximization is NP-hard, this procedure provides a $1 - 1/e$ approximation of the true optimum (Ko et al., 1995; Krause & Guestrin, 2012; Sharma et al., 2015).

### C.3 EFFECT OF REBALANCING ON DIVERSE SET OF HYPERPARAMETERS

While the WILDS datasets use a fixed set of hyperparameters proposed in the original paper (Koh et al., 2021), the standard setup for ColorMNIST in DomainBed benchmark prescribes a standard hyperparameter sweep (Gulrajani & Lopez-Paz, 2021). For each each environment (+90%, +80%, -90%) 20 hyperparameters sets are considered and three trials with different random seed are performed. This offers us an opportunity to investigate how rebalancing affects the model with various hyperparameters. In Figure 9 we show that rebalancing causes a favorable shift in the distribution of generalization performance over the different hyperparameters and trials.

The final performance in ColorMNIST is reported by choosing the best hyperparameter set in each of the three trials and averaging the final test error. Traditionally, the i.i.d. validation split is used to

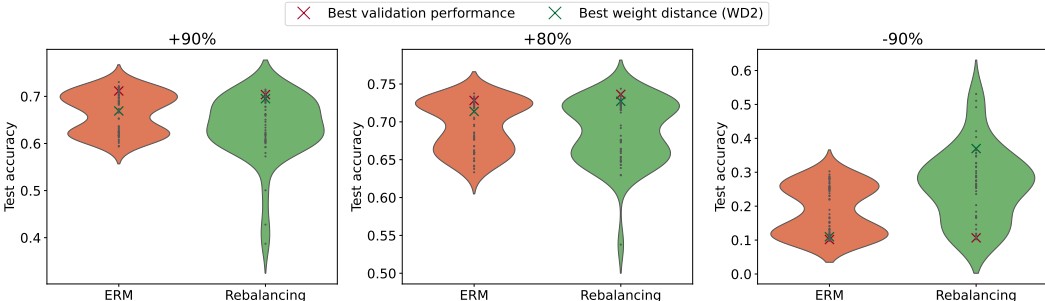

Figure 9: Results of all training runs conducted for ColorMNIST with no rebalancing and the best rebalancing (UMAP-8, label cond.). Results are reported for the final model for each training run (no early stopping). We can observe that rebalancing causes a noticeable shift in the performance of all hyperparameter sets, in the worst-case setting (-90% case). Here, we also observe, that choosing the hyperparameter set based on weight distance of the trained model to initialization results in a more favorable choice, compared to standard way of choosing the hyperparameter set using i.i.d. validation set.

determine the best hyperparameter set in each trial. But as outlined in DomainBed paper (Gulrajani & Lopez-Paz, 2021) model selection is key to achieve good generalization. In Figure C.3 we can see that using i.i.d. validation split to make this selection can lead to us selecting the worst checkpoints for generalization (-90% case). As discussed in Section 4.2, small weight distance of the trained model to initialization can be informative of potential for o.o.d. generalization. In Figure C.3 we indeed observe that selecting the hyperparameter set based on the weight distance at the end of model training results in us making a more favorable choice for o.o.d. generalization.

### C.4 COMBINING REBALANCING WITH OTHER DATA SMOOTHING APPROACHES

The main goal of rebalancing the training data is to more uniformly cover the data manifold, when training the model. Other approaches, such as mixup (Zhang, 2017) that interpolates data points in the training set, can also have a similar smoothing effect. As C-Mixup (Yao et al., 2022) achieves state of the art performance on PovertyMap dataset, it is natural to ask if these smoothing approaches can be combined. While C-Mixup was originally proposed for regression tasks and mixes datapoints based on their label similarity, we also adopted it to the categorical classification setup by mixing only points within the same class. We tested how combining C-Mixup with rebalancing would affect results on PovertyMap and iWildCam datasets. In all cases C-Mixup was used with CutMix (Yun et al., 2019) as proposed in the original paper for the PovertyMap task (Yao et al., 2022).

From Tables 4 and 5 we can see that combining C-Mixup and rebalancing tends to produce middle of the road results. In PovertyMap (Table 5) where C-Mixup is state of the art, the combined methods achieve similar results (up to experimental variance0. While for iWildCam (Table 4), where rebalancing is superior, the combined methods again perform worse than rebalancing alone but better than C-Mixup. This shows that the approaches can be used together but their benefits do not necessarily stack up.

### C.5 RESULTS ON DATASETS WITH POOR DENSITY FIT

As discussed in the main body, we do not hope to achieve great results when using sample reweighting if our density fit is poor or the test set features a domain shift. In this section we show the remaining WILDS (Koh et al., 2021) datasets we have considered that rely on pre-trained models. As can be seen in Figure 10 density fit quality is lacking, which translates in only small differences to vanilla ERM performance as seen in Tables 6, 7, 8 and 9. However, even with such poor density, rebalancing can occasionally help to improve worst-case o.o.d. performance, as seen in Table 7.

While our work shows the benefit of rebalancing, when we are able to fit a good density to the training data, further work is required to determine what embeddings should be used for each problem and how best to fit a density on those embeddings.

Table 4: *iWildCam*. Macro F1 and average classification accuracy (higher is better). o.o.d. results are on images from wildlife cameras not present in the training set, while i.d. results are on images from the cameras in the training set, but taken on different days. Parentheses show standard deviation across 3 replicates.

| Algorithm | Validation (i.d.) | | Validation (o.o.d.) | | Test (i.d.) | | Test (o.o.d.) | |
|---|---|---|---|---|---|---|---|---|
| | Macro F1 | Avg acc | Macro F1 | Avg acc | Macro F1 | Avg acc | Macro F1 | Avg acc |
| ERM | 48.8 (2.5) | 82.5 (0.8) | 37.4 (1.7) | 62.7 (2.4) | 47.0 (1.4) | 75.7 (0.3) | 31.0 (1.3) | 71.6 (2.5) |
| C-Mixup | 44.1 (0.8) | 80.5 (0.7) | 33.1 (0.6) | 57.2 (2.6) | 43.1 (0.9) | 71.9 (0.5) | 26.8 (1.4) | 70.2 (2.5) |
| Rebalancing | 49.1 (1.5) | 82.8 (1.7) | 38.8 (0.7) | 62.7 (0.6) | 48.1 (3.1) | 76.1 (1.0) | 31.5 (1.8) | 71.6 (1.2) |
| Rebalancing (PCA-256) | 51.4 (1.8) | 83.9 (1.3) | 39.7 (0.6) | 65.4 (0.5) | 47.0 (3.1) | 76.7 (1.2) | 33.5 (1.0) | 75.3 (0.6) |
| Rebalancing (label cond.) | 54.0 (1.5) | 84.5 (1.4) | 39.5 (0.5) | 66.7 (1.5) | 50.1 (1.8) | 77.5 (1.1) | 34.9 (1.1) | 77.4 (0.8) |
| Rebalancing (PCA, label cond.) | 53.9 (0.7) | 84.2 (0.4) | 40.2 (0.8) | 66.5 (1.5) | 49.8 (1.4) | 77.0 (0.3) | 35.5 (0.8) | 75.3 (2.8) |
| Rebalancing (C-Mixup) | 44.8 (0.9) | 80.9 (0.3) | 33.7 (0.6) | 58.8 (1.6) | 43.7 (2.0) | 72.0 (0.9) | 26.7 (0.3) | 70.6 (3.1) |
| Rebalancing (PCA-256, C-Mixup) | 44.1 (2.1) | 80.2 (1.7) | 34.3 (0.7) | 58.4 (0.9) | 42.6 (0.5) | 71.6 (0.6) | 27.7 (0.6) | 68.7 (1.6) |
| Rebalancing (label cond., C-Mixup) | 47.0 (1.9) | 80.8 (0.4) | 35.0 (0.3) | 58.8 (1.8) | 44.9 (1.7) | 72.6 (0.8) | 29.1 (0.9) | 69.7 (1.2) |
| Rebalancing (PCA, label cond., C-Mixup) | 50.3 (2.4) | 80.9 (1.0) | 35.3 (0.2) | 60.5 (2.7) | 44.3 (1.5) | 73.3 (0.6) | 31.3 (0.7) | 70.2 (0.2) |

Table 5: *PovertyMap*. Pearson correlation (higher is better) on in-distribution and out-of-distribution (unseen countries) held-out sets, incl. rural subpopulations. All results are averaged over 5 different o.o.d. country folds, with standard deviations across different folds in parentheses.

| Algorithm | Validation (i.d.) | | Validation (o.o.d.) | | Test (i.d.) | | Test (o.o.d.) | |
|---|---|---|---|---|---|---|---|---|
| | Overall | Worst | Overall | Worst | Overall | Worst | Overall | Worst |
| ERM | 0.82 (0.02) | 0.58 (0.07) | 0.80 (0.04) | 0.51 (0.06) | 0.82 (0.03) | 0.57 (0.07) | 0.78 (0.04) | 0.45 (0.06) |
| Rebalancing | 0.83 (0.01) | 0.62 (0.02) | 0.80 (0.03) | 0.53 (0.04) | 0.84 (0.02) | 0.63 (0.04) | 0.75 (0.07) | 0.44 (0.06) |
| Rebalancing (UMAP-64) | 0.85 (0.01) | 0.66 (0.03) | 0.80 (0.03) | 0.53 (0.04) | 0.85 (0.01) | 0.65 (0.04) | 0.78 (0.04) | 0.47 (0.10) |
| C-Mixup | 0.84 (0.01) | 0.64 (0.05) | 0.81 (0.04) | 0.55 (0.06) | 0.85 (0.01) | 0.64 (0.05) | 0.80 (0.04) | 0.51 (0.08) |
| Rebalancing (C-Mixup) | 0.83 (0.02) | 0.62 (0.09) | 0.79 (0.04) | 0.55 (0.05) | 0.84 (0.03) | 0.64 (0.07) | 0.79 (0.05) | 0.50 (0.06) |
| Rebalancing (C-Mixup, UMAP-64) | 0.84 (0.01) | 0.65 (0.04) | 0.82 (0.04) | 0.55 (0.05) | 0.85 (0.02) | 0.66 (0.05) | 0.79 (0.03) | 0.49 (0.07) |

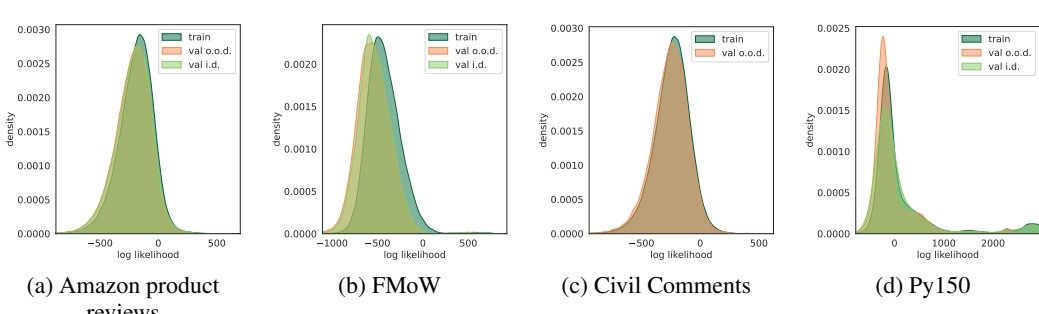

(a) Amazon product reviews    (b) FMoW    (c) Civil Comments    (d) Py150

Figure 10: Density fits with no dimensionality reduction for four WILDS (Koh et al., 2021) datasets, where the fit was poor.

Table 6: Baseline results on CivilComments. The reweighted (label) algorithm samples equally from the positive and negative class; the group DRO (label) algorithm additionally weights these classes so as to minimize the maximum of the average positive training loss and average negative training loss. We show standard deviation across 5 random seeds in parentheses.

| Algorithm | Avg val acc | Worst-group val acc | Avg test acc | Worst-group test acc |
|---|---|---|---|---|
| ERM | 92.3 (0.2) | 50.5 (1.9) | 92.2 (0.1) | 56.0 (3.6) |
| Reweighted (label) | 90.1 (0.4) | 65.9 (1.8) | 89.8 (0.4) | 69.2 (0.9) |
| Group DRO (label) | 90.4 (0.4) | 65.0 (3.8) | 90.2 (0.3) | 69.1 (1.8) |
| Rebalancing | 92.1 (0.3) | 49.7 (2.2) | 92.1 (0.28) | 55.7 (4.2) |
| Rebalancing (PCA-256) | 92.0 (0.2) | 49.9 (2.6) | 92.0 (0.17) | 55.8 (2.7) |
| Rebalancing (label cond.) | 90.1 (0.5) | 65.4 (2.7) | 89.9 (0.46) | 68.0 (2.2) |
| Rebalancing (PCA-256, label cond.) | 90.3 (0.3) | 65.7 (1.4) | 90.1 (0.34) | 69.4 (1.3) |

Table 7: Average and worst-region accuracies (%) under time shifts in FMoW. Models are trained on data before 2013 and tested on held-out location coordinates from i.d. and o.o.d. test sets. Parentheses show standard deviation across 3 replicates.

| Algorithm | Validation (i.d.) | | Validation (o.o.d.) | | Test (i.d.) | | Test (o.o.d.) | |
|---|---|---|---|---|---|---|---|---|
| | Overall | Worst | Overall | Worst | Overall | Worst | Overall | Worst |
| ERM | 61.2 (0.52) | 59.2 (0.69) | 59.5 (0.37) | 48.9 (0.62) | 59.7 (0.65) | 58.3 (0.92) | 53.0 (0.55) | 32.3 (1.25) |
| CORAL | 58.3 (0.28) | 55.9 (0.50) | 56.9 (0.25) | 47.1 (0.43) | 57.2 (0.90) | 55.0 (1.02) | 50.5 (0.36) | 31.7 (1.24) |
| IRM | 58.6 (0.07) | 56.6 (0.59) | 57.4 (0.37) | 47.5 (1.57) | 57.7 (0.10) | 56.0 (0.34) | 50.8 (0.13) | 30.0 (1.37) |
| Group DRO | 60.5 (0.36) | 57.9 (0.62) | 58.8 (0.19) | 46.5 (0.25) | 59.4 (0.11) | 57.8 (0.60) | 52.1 (0.50) | 30.8 (0.81) |
| Rebalancing | 60.5 (0.54) | 58.4 (0.73) | 58.6 (0.57) | 50.9 (1.52) | 59.0 (0.36) | 57.4 (0.86) | 52.4 (0.57) | 33.6 (1.23) |
| Rebalancing (PCA-256) | 60.9 (0.50) | 58.5 (0.51) | 58.9 (0.57) | 52.0 (1.01) | 59.6 (0.34) | 57.8 (0.88) | 52.7 (0.79) | 33.7 (0.87) |

Table 8: *Amazon product reviews.* We report the accuracy of models trained using ERM, CORAL, IRM, and group DRO, as well as a reweighting baseline that reweights for class balance. To measure tail performance across reviewers, we also report the accuracy for the reviewer in the 10th percentile.

| Algorithm | Validation (i.d.) | | Validation (o.o.d.) | | Test (i.d.) | | Test (o.o.d.) | |
|---|---|---|---|---|---|---|---|---|
| | 10th percentile | Average | 10th percentile | Average | 10th percentile | Average | 10th percentile | Average |
| ERM | 58.7 (0.0) | 75.7 (0.2) | 55.2 (0.7) | 72.7 (0.1) | 57.3 (0.0) | 74.7 (0.1) | 53.8 (0.8) | 71.9 (0.1) |
| CORAL | 56.2 (1.7) | 74.4 (0.3) | 54.7 (0.0) | 72.0 (0.3) | 55.1 (0.4) | 73.4 (0.2) | 52.9 (0.8) | 71.1 (0.3) |
| IRM | 56.4 (0.8) | 74.3 (0.1) | 54.2 (0.8) | 71.5 (0.3) | 54.7 (0.8) | 72.9 (0.2) | 52.4 (0.8) | 70.5 (0.3) |
| Group DRO | 57.8 (0.8) | 73.7 (0.6) | 54.7 (0.0) | 70.7 (0.6) | 55.8 (1.0) | 72.5 (0.3) | 53.3 (0.0) | 70.0 (0.5) |
| Label reweighted | 55.1 (0.8) | 71.9 (0.4) | 52.1 (0.2) | 69.1 (0.5) | 54.4 (0.4) | 70.7 (0.4) | 52.0 (0.0) | 68.6 (0.6) |
| Rebalancing | 57.8 (0.8) | 75.0 (0.4) | 54.7 (0.0) | 71.9 (0.3) | 57.8 (0.8) | 73.7 (0.3) | 53.8 (0.8) | 71.2 (0.3) |
| Rebalancing (PCA-256) | 57.8 (0.8) | 75.2 (0.1) | 55.1 (0.8) | 72.1 (0.0) | 57.8 (0.8) | 73.9 (0.1) | 53.3 (0.0) | 71.2 (0.1) |

# D    RELATION BETWEEN DD AND EXPECTED RISKS

The relation between DD and expected risks is subtle. Clearly, the DD risk is always larger than the expected risk, however, the gap between the two can be zero when considering the optimal learner:

**Theorem D.1.** *For any loss $l : \mathbb{Y} \times \mathbb{Y} \to \mathbb{R}_+$ that is convex w.r.t. its first argument, we have*

$$\max_{q \in \mathcal{Q}_\gamma} \min_{f \in \mathcal{F}} r_{exp}(f; q) = \min_{f \in \mathcal{F}} r_{dd}(f; \gamma) \,,$$

*given that $\mathcal{F}$ is a compact convex set and all densities $q : \mathcal{X} \to [0, 1]$ are defined over a bounded domain.*

This is a somewhat abstract result that we include mainly for completeness. We find its usefulness only as a confirmation that the optimal distribution DD risk will not be worse than the best expected risk over the worst distribution in $\mathcal{Q}_\gamma$.

## D.1    PROOF THEOREM D.1

*Proof.* It follows from the max–min inequality, that the following relation generally holds:

$$\max_{q \in \mathcal{Q}_\gamma} \min_{f \in \mathcal{F}} r_{\exp}(f; q) \le \min_{f \in \mathcal{F}} \max_{q \in \mathcal{Q}_\gamma} r_{\exp}(f; q) = \min_{f \in \mathcal{F}} r_{\mathrm{dd}}(f; \gamma).$$

It is a consequence of Sion's minimax theorem (Komiya, 1988) that the above is an equality when $r_{\exp}(f; q)$ is quasi-convex w.r.t. $f$, quasi-concave w.r.t. $q$, and when both $\mathcal{F}$ and $\mathcal{Q}_\gamma$ are compact convex sets.

**Convexity w.r.t. $f$ and concavity w.r.t. $q$.** We know that the expected risk is linear w.r.t $q$ as

$$r_{\exp}(f; q) = \int_{\mathcal{X}} q(x)\, l\big(f(x), f^*(x)\big)\, dx \,.$$

Further, if we assume that the loss function is convex w.r.t. $f$ then the expected risk is also convex, since the convex combination of convex functions is also convex (see Appendix D.2).

**Convexity of $\mathcal{Q}_\gamma$.** Let $\Omega$ be a bounded domain in $\mathbb{R}^d$. The set of distributions we are interested in is those that have entropy at least $\gamma + H(u)$. The entropy $H(p)$ of a probability distribution $p$ over

Table 9: Results on Py150. We report both the model's accuracy on predicting class and method tokens and accuracy on all tokens trained using ERM, CORAL, IRM and group DRO. Standard deviations over 3 trials are in parentheses.

| Algorithm | Validation (i.d.) | | Validation (o.o.d.) | | Test (i.d.) | | Test (o.o.d.) | |
|---|---|---|---|---|---|---|---|---|
| | Method/class | All | Method/class | All | Method/class | All | Method/class | All |
| ERM | 75.5 (0.5) | 74.6 (0.4) | 68.0 (0.1) | 69.4 (0.1) | 75.4 (0.4) | 74.5 (0.4) | 67.9 (0.1) | 69.6 (0.1) |
| CORAL | 70.7 (0.0) | 70.9 (0.1) | 65.7 (0.2) | 67.2 (0.1) | 70.6 (0.0) | 70.8 (0.1) | 65.9 (0.1) | 67.9 (0.0) |
| IRM | 67.3 (1.1) | 68.4 (0.7) | 63.9 (0.3) | 65.6 (0.1) | 67.3(1.1) | 68.3 (0.7) | 64.3 (0.2) | 66.4 (0.1) |
| Group DRO | 70.8 (0.0) | 71.2 (0.1) | 65.4 (0.0) | 67.3 (0.0) | 70.8 (0.0) | 71.0 (0.0) | 65.9 (0.1) | 67.9 (0.0) |
| Rebalancing | 75.1 (0.5) | 74.4 (0.4) | 67.0 (0.1) | 69.0 (0.2) | 74.9 (0.5) | 74.2 (0.4) | 67.2 (0.1) | 69.2 (0.2) |
| Rebalancing (PCA-256) | 75.2 (0.5) | 74.3 (0.4) | 67.1 (0.1) | 69.0 (0.1) | 75.0 (0.5) | 74.2 (0.4) | 67.2 (0.0) | 69.1 (0.1) |

a bounded domain $\Omega$ is a concave function. For any two distributions $p$ and $q$ on $\Omega$, and for any $\lambda \in [0, 1]$, the entropy $H$ satisfies

$$H\big(\lambda p + (1 - \lambda)q\big) \geq \lambda H(p) + (1 - \lambda)H(q).$$

Given this property, if $p$ and $q$ are in the set $\mathcal{Q}_\gamma$ (i.e., $H(p) \geq \gamma + H(u)$ and $H(q) \geq \gamma + H(u)$), then for any $\lambda \in [0, 1]$,

$$H(\lambda p + (1 - \lambda)q) \geq \lambda H(p) + (1 - \lambda)H(q) \geq \lambda\big(\gamma + H(u)\big) + (1 - \lambda)\big(\gamma + H(u)\big) = \gamma + H(u).$$

Thus, $\lambda p + (1 - \lambda)q \in \mathcal{Q}_\gamma$, demonstrating that $\mathcal{Q}_\gamma$ is a convex set.

**Compactness.** In the space of probability distributions on a bounded domain $\Omega$, the set of all distributions is bounded because the total probability mass is 1, and $\Omega$ itself is bounded. We need to check whether $\mathcal{Q}_\gamma$ is closed in the weak topology. In the space of probability distributions, a sequence of distributions $\{p_n\}$ converges weakly to $p$ if for all bounded continuous functions $f$,

$$\lim_{n \to \infty} \int f \, dp_n = \int f \, dp.$$

Entropy is lower semi-continuous in the weak topology, which means that if a sequence of distributions $\{p_n\}$ converges weakly to $p$, then $\liminf_{n \to \infty} H(p_n) \geq H(p)$.

To show that $\mathcal{Q}_\gamma$ is closed, suppose we have a sequence $\{p_n\}$ in $q$ such that $p_n \to p$ weakly. Since $p_n \in \mathcal{Q}_\gamma$, we have $H(p_n) \geq k$ for all $n$. Using the lower semi-continuity of entropy:

$$H(p) \geq \liminf_{n \to \infty} H(p_n) \geq k.$$

Therefore, $p \in \mathcal{Q}_\gamma$, showing that $\mathcal{Q}_\gamma$ is closed in the weak topology.

Since $\mathcal{Q}_\gamma$ is convex, closed, and bounded in the space of probability distributions on a bounded domain $\Omega$, we can conclude that $\mathcal{Q}_\gamma$ is a compact convex set. $\square$

### D.2 PROOF OF CONVEXITY OF $r_{\text{EXP}}(f; p)$ W.R.T. THE FIRST ARGUMENT

Let $r_{\exp}(f; p) = \sum_x p(x)l(f(x), f^*(x))$, where $l$ is convex with respect to $f(x)$. We aim to show that $r_{\exp}(f; p)$ is convex with respect to the function $f$.

Consider any two functions $f_1$ and $f_2$ and a scalar $\lambda \in [0, 1]$. We need to show that:

$$r_{\exp}(\lambda f_1 + (1 - \lambda)f_2; p) \leq \lambda r_{\exp}(f_1; p) + (1 - \lambda)r_{\exp}(f_2; p).$$

First, express $r_{\exp}(\lambda f_1 + (1 - \lambda)f_2; p)$:

$$r_{\exp}(\lambda f_1 + (1 - \lambda)f_2; p) = \sum_x p(x)l((\lambda f_1 + (1 - \lambda)f_2)(x), f^*(x)).$$

Since $(\lambda f_1 + (1 - \lambda)f_2)(x) = \lambda f_1(x) + (1 - \lambda)f_2(x)$, we have:

$$r_{\exp}(\lambda f_1 + (1 - \lambda)f_2; p) = \sum_x p(x)l(\lambda f_1(x) + (1 - \lambda)f_2(x), f^*(x)).$$

By the convexity of $l$ in its first argument, we have

$$l(\lambda f_1(x) + (1 - \lambda)f_2(x), f^*(x)) \leq \lambda l(f_1(x), f^*(x)) + (1 - \lambda)l(f_2(x), f^*(x)).$$

Multiplying both sides by $p(x)$ and summing over $x$ gives,

$$\sum_x p(x)l(\lambda f_1(x) + (1 - \lambda)f_2(x), f^*(x)) \leq \sum_x p(x)(\lambda l(f_1(x), y) + (1 - \lambda)l(f_2(x), f^*(x))),$$

while distributing the sums leads to

$$\sum_x p(x)(\lambda l(f_1(x), f^*(x)) + (1 - \lambda)l(f_2(x), f^*(x))) = \lambda \sum_x p(x)l(f_1(x), f^*(x))$$
$$+ (1 - \lambda) \sum_x p(x)l(f_2(x), f^*(x)).$$

We have thus far shown that

$$r_{\exp}(\lambda f_1 + (1 - \lambda)f_2; p) \leq \lambda r_{\exp}(f_1; p) + (1 - \lambda)r_{\exp}(f_2; p).$$

Since the above inequality holds for any $f_1, f_2$, and $\lambda \in [0, 1]$, $r_{\exp}(f; p)$ is convex with respect to $f$. This concludes the proof.

## E  PROOF OF THEOREM 3.1

**Theorem 3.1.** *Consider a zero-one loss and suppose that we can train a classifier up to some fixed expected risk $\varepsilon < 1/2$ under any distribution. A classifier optimized for the uniform distribution will yield the smallest DD risk:*

$$\max_{f \in \mathcal{F}_{u,\varepsilon}} r_{dd}(f; \gamma) \leq \max_{f \in \mathcal{F}_{p,\varepsilon}} r_{dd}(f; \gamma) \quad \text{for all} \quad p \neq u. \tag{6}$$

*Proof.* The theorem is asking the following question: assuming we learn a classifier w.r.t some training distribution $p$, and we know that the expected risk w.r.t. $p$ is $\varepsilon$, what is the worst case expected risk over all possible data distributions $q \in \mathcal{Q}_\gamma$ and learned functions $f$? To answer this question, we first show that the DD risk grows proportionally with the volume of the set of examples $\mathcal{E} \subset \mathcal{X}$ that the classifier mislabels, and then argue that the worst classifier within $\mathcal{F}_{u,\varepsilon}$ has smaller such volume as compared to the worst classifier within some $\mathcal{F}_{p,\varepsilon}$ where $p \neq u$.

Denote by $\mathcal{E} \subset \mathcal{X}$ the set which contains all instances that a classifier $f$ mislabels:

$$\mathcal{E} = \{x \in \mathcal{X} \text{ such that } f(x) \neq f^*(x)\}$$

Our first step in proving the theorem entails characterizing the relation between DD risk and error volume $\text{vol}(\mathcal{E})$. To that end, the following lemma characterizes the distribution $q^* \in \mathcal{Q}_\gamma$ that maximizes (3):

**Lemma E.1.** *Amongst all densities with $q(\mathcal{E}) = \epsilon$ the one that has the maximum entropy is given by*

$$q_\epsilon^*(x) = \begin{cases} \epsilon/vol(\mathcal{E}) & x \in \mathcal{E} \\ (1 - \epsilon)/vol(\mathcal{X} - \mathcal{E}) & otherwise, \end{cases}$$

*achieving entropy of $H(q_\epsilon^*) = \epsilon \left(\log(vol(\mathcal{E})) - \log(\epsilon)\right) + (1 - \epsilon)\left(\log(vol(\mathcal{X} - \mathcal{E})) - \log(1 - \epsilon)\right)$. Furthermore, if $\gamma$ is chosen such that $\epsilon$ is the maximal value satisfying $q_\epsilon^* \in \mathcal{Q}_\gamma$, the DD risk is given by $r_{dd}(f; \gamma) = \epsilon$.*

The proof of the Lemma is provided in Appendix E.1.

For any fixed $\gamma$, the worst-case distribution $q^*$ (equivalently, the distribution $q_\epsilon^* \in \mathcal{Q}_\gamma$ with the largest $\epsilon$) will be piece-wise uniform in $\mathcal{E}$ and $\mathcal{X} - \mathcal{E}$, respectively, and the DD risk is exactly equal to $q^*(\mathcal{E}) = \epsilon$. The DD risk is smaller than one when the mislabeled set is not sufficiently large to satisfy the entropy lower bound $H(u) - \gamma$. In those cases, $q^*$ will assign the maximum probability density to $q^*(\mathcal{E})$ while ensuring that the entropy lower bound is met. Notice that, for any fixed $\epsilon$, entropy is a monotonically increasing function of $\text{vol}(\mathcal{E})$ (since by assumption $\text{vol}(\mathcal{E}) \leq \text{vol}(\mathcal{X} - \mathcal{E})$). As such, the DD risk of a classifier $f$ grows with the error volume.

With this in place, to prove the theorem it suffices to show that $\forall p \neq u \; \mathcal{F}_{p,\varepsilon}$ always contains a function whose error volume is greater than $\varepsilon$. More formally, there exists $f \in \mathcal{F}_{p,\varepsilon}$ such that

$\text{vol}(\mathcal{E}_p) = \int_X \ell\big(f(x), f^*(x)\big)\mathrm{d}x > \varepsilon$. Showing this suffices because it follows from the definition that $\text{vol}(\mathcal{E}_u) = \int_X \ell\big(f(x), f^*(x)\big)\mathrm{d}x = \varepsilon$ for every $f \in \mathcal{F}_{u,\varepsilon}$.

The aforementioned claim follows by noting that since $p$ is non-uniform, there exists a region of volume strictly greater than $\varepsilon$ whose mass under $p$ is exactly $\varepsilon$. If not, either $p$ is the uniform distribution or it cannot be a valid probability distribution as it must have a total probability mass above 1. Then, we can always find some $f \in \mathcal{F}_{p,\varepsilon}$ so that $\ell\big(f(x), f^*(x)\big)$ is 1 in this region and 0 elsewhere. This concludes our argument. $\qquad\square$

### E.1    PROOF OF LEMMA E.1

We repeat the lemma setup and statement here for completeness:

Denote by $\mathcal{E} \subset \mathcal{X}$ the set which contains all instances that a classifier $f$ miss-labels:

$$\mathcal{E} = \{x \in \mathcal{X} \text{ such that } f(x) \neq f^*(x)\}$$

**Lemma E.1.** *Amongst all densities with $q(\mathcal{E}) = \epsilon$ the one that has the maximum entropy is given by*

$$q_\epsilon^*(x) = \begin{cases} \epsilon/vol(\mathcal{E}) & x \in \mathcal{E} \\ (1-\epsilon)/vol(\mathcal{X} - \mathcal{E}) & otherwise, \end{cases}$$

*achieving entropy of $H(q_\epsilon^*) = \epsilon\left(\log(vol(\mathcal{E})) - \log(\epsilon)\right) + (1-\epsilon)\left(\log(vol(\mathcal{X} - \mathcal{E})) - \log(1-\epsilon)\right)$. Furthermore, if $\gamma$ is chosen such that $\epsilon$ is the maximal value satisfying $q_\epsilon^* \in \mathcal{Q}_\gamma$, the DD risk is given by $r_{dd}(f; \gamma) = \epsilon$.*

*Proof.* The distributionally diverse risk of $f$ can be determined by identifying the distribution $q^* \in \mathcal{Q}_\gamma$ that maximizes $q^*(\mathcal{E})$. Consider any partitioning of $\mathcal{X}$ into sets $\mathcal{E}$ and $\mathcal{E}^\perp = \mathcal{X} - \mathcal{E}$. We claim that, amongst all densities with $q(\mathcal{E}) = \epsilon$ (and thus the same DD risk) the one that has the maximum entropy is given by

$$q_\epsilon^*(x) = \begin{cases} \epsilon/\text{vol}(\mathcal{E}) & x \in \mathcal{E} \\ (1-\epsilon)/\text{vol}(\mathcal{E}^\perp) & \text{otherwise,} \end{cases}$$

achieving entropy of

$$H(q_\epsilon^*) = \int_{\mathcal{E}} \frac{\epsilon}{\text{vol}(\mathcal{E})} \log(\text{vol}(\mathcal{E})/\epsilon)dx + \int_{\mathcal{E}^\perp} \frac{1-\epsilon}{\text{vol}(\mathcal{E}^\perp)} \log(\text{vol}(\mathcal{E}^\perp)/(1-\epsilon))dx$$

$$= \epsilon\left(\log(\text{vol}(\mathcal{E})) - \log(\epsilon)\right) + (1-\epsilon)\left(\log(\text{vol}(\mathcal{E}^\perp)) - \log(1-\epsilon)\right).$$

To see this, we notice that the entropy over $\mathcal{X}$ can be decomposed in terms of the entropy of the conditional distributions defined on $\mathcal{E}$ and $\mathcal{E}^\perp$:

$$H(q) = \int_{\mathcal{E}} q(x) \log(1/q(x))dx + \int_{\mathcal{E}^\perp} q(x) \log(1/q(x))dx$$

$$= q(\mathcal{E}) \int_{\mathcal{E}} \frac{q(x)}{q(\mathcal{E})} \log(\frac{q(\mathcal{E})}{q(x)q(\mathcal{E})})dx + q(\mathcal{E}^\perp) \int_{\mathcal{E}^\perp} \frac{q(x)}{q(\mathcal{E}^\perp)} \log(\frac{q(\mathcal{E}^\perp)}{q(x)q(\mathcal{E}^\perp)})dx$$

$$= \epsilon \int_{\mathcal{E}} q_{\mathcal{E}}(x) \log(\frac{1}{q_{\mathcal{E}}(x)q(\mathcal{E})})dx + (1-\epsilon) \int_{\mathcal{E}^\perp} q_{\mathcal{E}^\perp}(x) \log(\frac{1}{q_{\mathcal{E}^\perp}(x)q(\mathcal{E}^\perp)})dx \qquad (*)$$

$$= \epsilon\left(H(q_{\mathcal{E}}) - \log(\epsilon)\right) + (1-\epsilon)\left(H(q_{\mathcal{E}^\perp}) - \log(1-\epsilon)\right)$$

Where in $(*)$ we define $q_{\mathcal{E}}(x) = q(x)/q(\mathcal{E})$ and $q_{\mathcal{E}^\perp}(x) = q(x)/q(\mathcal{E}^\perp)$ to be densities supported in $\mathcal{E}$ and $\mathcal{E}^\perp$, respectively.

It is well known that the maximal entropy for distributions supported on a set $\mathcal{E}$ is achieved by the uniform distribution and is given by $\max_{q_{\mathcal{E}}} H(q_{\mathcal{E}}) = -\int_{\mathcal{E}} q_{\mathcal{E}}(\mathcal{E}) \log(q_{\mathcal{E}}(\mathcal{E}))dx = \log(\text{vol}(\mathcal{E}))$ (and similarly for $q_{\mathcal{E}^\perp}$, respectively), from which it follows that

$$H(q) \leq \epsilon\left(\log(\text{vol}(\mathcal{E})) - \log(\epsilon)\right) + (1-\epsilon)\left(\log(\text{vol}(\mathcal{E}^\perp)) - \log(1-\epsilon)\right) = H(q_\epsilon^*),$$

meaning that the upper bound is exactly achieved by $q_\epsilon^*$.

It now easily follows that $r_{\mathrm{dd}}(f; \gamma) = \epsilon$. Note that in the case of the zero-one loss, the expectation in the definition of $r_{\mathrm{dd}}$ is simply $q(\mathcal{E}) = \epsilon$. Since $q_\epsilon^* \in \mathcal{Q}_\gamma$, we immediately have $r_{\mathrm{dd}}(f; \gamma) \geq \epsilon$. The reverse inequality, $r_{\mathrm{dd}}(f; \gamma) \leq \epsilon$, follows from the maximality of $\epsilon$. Otherwise, we would have some $q' \in \mathcal{Q}_\gamma$ with $q'(\mathcal{E}) = \epsilon' > \epsilon$. But by the first part of the theorem, this would imply that $q_{\epsilon'}^* \in \mathcal{Q}_\gamma$, contradicting the maximality of $\epsilon$. $\qquad\square$

## F  PROOF OF THEOREM 3.2

**Theorem 3.2.** *The DD risk of a classifier under the zero-one loss is at most*

$$r_{dd}(f; \gamma) \leq \min \left\{ \frac{\gamma - \log\left(\frac{1-\alpha}{1-r_{exp}(f;u)}\right)}{\log\left(\frac{\alpha}{1-\alpha}\right) + \log\left(\frac{1}{r_{exp}(f;u)} - 1\right)}, \; r_{exp}(f; u) + \sqrt{\frac{\gamma}{2}} \right\},$$

*where $\alpha \in (r_{exp}(f; u), 1)$ may be chosen freely. The DD risk is below 1 for $r_{exp}(f; u) < e^{-\gamma}$.*

*Proof.* To characterize the DD risk $r_{\mathrm{dd}}(f; \gamma)$, we consider the density defined in Lemma E.1:

$$q_\epsilon^*(x) = \begin{cases} \epsilon/\mathrm{vol}(\mathcal{E}) & x \in \mathcal{E} \\ (1-\epsilon)/\mathrm{vol}(\mathcal{X} - \mathcal{E}) & \text{otherwise} \end{cases},$$

and proceed to identify the largest $\epsilon$ such that $q_\epsilon^* \in \mathcal{Q}_\gamma$:

$$r_{\mathrm{dd}}(f; \gamma) = \max_{\epsilon \leq 1} \epsilon \quad \text{subject to} \quad H(q_\epsilon^*) \geq H(u) - \gamma.$$

We are interested in the regime $\log(\mathrm{vol}(\mathcal{E})) < H(u) - \gamma$ or equivalently $\mathrm{vol}(\mathcal{E}/\mathcal{X}) = r_{\exp}(f; u) < \exp(-\gamma)$. When the above inequality is not met, the DD risk is trivially 1 as the entropy constraint is not violated for any $\epsilon$.

To proceed, we notice that the entropy can be written as a KL-divergence between a Bernoulli random variable $v_1$ that is equal to one with probability $P(v_1 = 1) = \epsilon$ and a Bernoulli random variable $v_2$ that is equal to one with probability $P(v_2 = 1) = r_{\exp}(f; u)$ and zero with probability $P(v_2 = 0) = 1 - r_{\exp}(f; u)$.

$$\begin{aligned}
H(q_\epsilon^*) &= \epsilon \left( \log\left(\frac{\mathrm{vol}(\mathcal{E})}{\mathrm{vol}(\mathcal{X})}\right) + \log(\mathrm{vol}(\mathcal{X})) - \log(\epsilon) \right) \\
&\quad + (1-\epsilon)\left( \log\left(\frac{\mathrm{vol}(\mathcal{X}) - \mathrm{vol}(\mathcal{E})}{\mathrm{vol}(\mathcal{X})}\right) + \log(\mathrm{vol}(\mathcal{X})) - \log(1-\epsilon) \right) \\
&= \epsilon\left(\log(r_{\exp}(f;u)) - \log(\epsilon)\right) + (1-\epsilon)\left(\log(1 - r_{\exp}(f;u)) - \log(1-\epsilon)\right) + \log(\mathrm{vol}(\mathcal{X})) \\
&= \epsilon \log\left(\frac{r_{\exp}(f;u)}{\epsilon}\right) + (1-\epsilon)\log\left(\frac{1 - r_{\exp}(f;u)}{1-\epsilon}\right) + \log(\mathrm{vol}(\mathcal{X})) \\
&= \log(\mathrm{vol}(\mathcal{X})) - D_{\mathrm{KL}}(v_1 \| v_2) \\
&= H(u) - D_{\mathrm{KL}}(v_1 \| v_2)
\end{aligned}$$

or equivalently $D_{\mathrm{KL}}(v_1 \| v_2) = H(u) - H(q_\epsilon^*)$. We will derive two alternative bounds, each of which is tight in a different regime.

For the first bound, we rely on Pinsker's inequality:

$$\begin{aligned}
D_{\mathrm{KL}}(v_1 \| v_2) &\geq \left(|\epsilon - r_{\exp}(f; u)| + |(1 - \epsilon - (1 - r_{\exp}(f; u)))|\right)^2 / 2 \\
&= 2|\epsilon - r_{\exp}(f; u)|^2
\end{aligned}$$

implying

$$\sqrt{\frac{H(u) - H(q_\epsilon^*)}{2}} \geq |\epsilon - r_{\exp}(f; u)|.$$

We now take $\epsilon$ to be maximal, i.e., $\epsilon = r_{\mathrm{dd}}(f; \gamma)$, and apply the constraint $H(u) - H(q_\epsilon^*) \leq \gamma$ to obtain

$$|r_{\mathrm{dd}}(f; \gamma) - r_{\exp}(f; u)| \leq \sqrt{\frac{\gamma}{2}}.$$

By definition of the DD risk, we note that the optimal choice of $\epsilon$ must abide to $\epsilon \geq r_{\exp}(f; u)$ (otherwise, $u$ leads to a worse expected error than $q_\epsilon^*$), meaning

$$r_{dd}(f; \gamma) \leq r_{\exp}(f; u) + \sqrt{\frac{\gamma}{2}},$$

which shows that the gap converges to zero as $\gamma \to 0$, but wrongly suggests that the DD risk is never below $\sqrt{\frac{\gamma}{2}}$ even when $r_{\exp}(f; u) = 0$.

We may also derive a tighter (and more involved) bound if we take a Taylor-series expansion of $f(\epsilon) = D_{\mathrm{KL}}(v_1 \| v_2)$ at $\epsilon = \alpha \in (r_{\exp}(f; u), 1)$:

$$f(\epsilon) \geq f(\alpha) + f'(\alpha)(\epsilon - \alpha)$$

$$= \epsilon \left( \log \left( \frac{\alpha}{1-\alpha} \right) + \log \left( \frac{1}{r_{\exp}(f; u)} - 1 \right) \right) + \log \left( \frac{1-\alpha}{1 - r_{\exp}(f; u)} \right)$$

Substituting $D_{\mathrm{KL}}(v_1 \| v_2) \leq \gamma$ as above and solving for $\epsilon$ then yields:

$$\gamma - \log \left( \frac{1-\alpha}{1 - r_{\exp}(f; u)} \right) \geq \epsilon \left( \log \left( \frac{\alpha}{1-\alpha} \right) + \log \left( \frac{1}{r_{\exp}(f; u)} - 1 \right) \right).$$

implying

$$r_{dd}(f; \gamma) \leq \min_{a \in (r_{\exp}(f; u), 1)} \left\{ \frac{\gamma - \log \left( \frac{1-\alpha}{1-r_{\exp}(f; u)} \right)}{\log \left( \frac{\alpha}{1-\alpha} \right) + \log \left( \frac{1}{r_{\exp}(f; u)} - 1 \right)} \right\}.$$

For our simplified bound, we will utilize the following inequality involving the KL-divergence:

$$D_{\mathrm{KL}}(v_1 \| v_2) \geq \epsilon \log \left( \frac{1}{r_{\exp}(f; u)} \right) - \log(2)$$

which can be derived by a Taylor-series expansion of $f(\epsilon) = D_{\mathrm{KL}}(v_1 \| v_2)$ at $\epsilon = 0.5$:

$$f(\epsilon) \geq f\left(\frac{1}{2}\right) + f'\left(\frac{1}{2}\right)\left(\epsilon - \frac{1}{2}\right)$$

$$= \epsilon \log \left( \frac{1}{r_{\exp}(f; u)} \right) - \log(2) + (1 - \epsilon) \log \left( \frac{1}{1 - r_{\exp}(f; u)} \right)$$

$$\geq \epsilon \log \left( \frac{1}{r_{\exp}(f; u)} \right) - \log(2)$$

Substituting $D_{\mathrm{KL}}(v_1 \| v_2) \leq \gamma$ as above and solving for $\epsilon$ then yields:

$$\gamma + \log(2) \geq \epsilon \log \left( \frac{1}{r_{\exp}(f; u)} \right).$$

We thus have

$$r_{dd}(f; \gamma) \leq \frac{\gamma + \log(2)}{\log \left( \frac{1}{r_{\exp}(f; u)} \right)},$$

which reveals that, as expected, the DD risk converges to zero as $r_{\exp}(f; u) \to 0$ for any $\gamma$. $\qquad \square$

## G   PROOF OF THEOREM 4.1

**Theorem 4.1.** *For any unbiased prior $\pi$, the DD risk of a stochastic learner is at most*

$$r_{dd}(\pi_Z; \gamma) := \max_{q \in \mathcal{Q}_\gamma} \mathbb{E}_{f \sim \pi_Z}[r_{exp}(f, q)] \leq \mathbb{E}_{f \sim \pi_Z}[r_{exp}(f, p_Z)] + 2\,\delta(\pi_Z, \pi),$$

*where $l$ is a loss such that $\sum_{y \in \mathcal{Y}} l(y, y') = \sum_{y \in \mathcal{Y}} l(y, y'') \; \forall y', y'' \in \mathcal{Y}$, such as the zero-one loss.*

*Proof.* The expected risk difference is given by:

$$|r(\pi_Z, q) - r(\pi_Z, p_Z)| = |\sum_{f \in \mathcal{F}} \pi_Z(f) r_{\exp}(f, q) - \sum_{f \in \mathcal{F}} \pi_Z(f) r_{\exp}(f, p_Z)|$$

$$= |\sum_{f \in \mathcal{F}} (\pi_Z(f) - \pi(f)) \, r_{\exp}(f, q) - \sum_{f \in \mathcal{F}} (\pi_Z(f) - \pi(f)) \, r_{\exp}(f, p_Z)$$

$$+ \sum_{f \in \mathcal{F}} \pi(f) \, (r_{\exp}(f, p_Z) - r(f, q)) \, |$$

$$\leq \sum_{f \in \mathcal{F}} |\pi_Z(f) - \pi(f)| \, (\sup_{f \in \mathcal{F}} r_{\exp}(f, q) + \sup_{f \in \mathcal{F}} r_{\exp}(f, p_Z))$$

$$+ |\sum_{f \in \mathcal{F}} \pi(f) \, (r_{\exp}(f, p_Z) - r_{\exp}(f, q)) \, |$$

$$= 2 \, \delta(\pi_Z, \pi) + |\mathbb{E}_{f \sim \pi}[r_{\exp}(f, p_Z)] - \mathbb{E}_{f \sim \pi}[r_{\exp}(f, q)]|$$

$$= 2 \, \delta(\pi_Z, \pi) + |r_{\exp}(\pi, p_Z) - r_{\exp}(\pi, q)|$$

implying

$$\mathbb{E}_{f \sim \pi_Z}[r_{\exp}(f, q)] \leq \mathbb{E}_{f \sim \pi_Z}[r_{\exp}(f, p_Z)] + 2 \, \delta(\pi_Z, \pi) + |r_{\exp}(\pi, p_Z) - r_{\exp}(\pi, q)|$$

Taking the maximum over all distributions $q$ of sufficient entropy yields:

$$\max_{q \in \mathcal{Q}_\gamma} \mathbb{E}_{f \sim \pi_Z}[r_{\exp}(f, q)] \leq \mathbb{E}_{f \sim \pi_Z}[r_{\exp}(f, p_Z)] + 2 \, \delta(\pi_Z, \pi) + \max_{q \in \mathcal{Q}_\gamma} |r_{\exp}(\pi, p_Z) - r_{\exp}(\pi, q)|$$

If we select $\pi$ to be an uninformed prior and our hypothesis class to sufficiently diverse, we expect the right-most term to be negligible for typical distributions.

More precisely, in the standard case the term is zero if the prior assigns to every possible data labeling the same probability. Formally, we say that a prior $\pi$ over the hypothesis class is *unbiased* if, for any $x$ and every $y$, we have $\pi(f(x) = y) = 1/|\mathcal{Y}|$. In this case, the right-most term vanishes as

$$\mathbb{E}_{f \sim \pi}[r_{\exp}(f, q)] = \sum_f \pi(f) \sum_z q(z) \, l \, (f(x), f^*(x))$$

$$= \sum_z q(z) \sum_f \pi(f) \, l \, (f(x), f^*(x))$$

$$= \sum_z q(z) \left( \frac{1}{|\mathcal{Y}|} \sum_{y \in \mathcal{Y}} l \, (y, f^*(x)) \right) \qquad \text{(unbiased } \pi\text{)}$$

$$= c \qquad \text{(balanced loss)}$$

$$= \mathbb{E}_{f \sim \pi}[r_{\exp}(f, p_Z)]$$

In the last step, we assumed that the loss is balanced such that for any $y', y'' \in \mathcal{Y}$:

$$\sum_{y \in \mathcal{Y}} l \, (y, y') = \sum_{y \in \mathcal{Y}} l \, (y, y'') \qquad (10)$$

Thus, the error induced by covariate shift reduces to the distance between prior and posterior:

$$r_{\text{dd}}(\pi_Z; \gamma) := \max_{q \in \mathcal{Q}_\gamma} \mathbb{E}_{f \sim \pi_Z}[r_{\exp}(f, q)] \leq \mathbb{E}_{f \sim \pi_Z}[r_{\exp}(f, p_Z)] + 2 \, \delta(\pi_Z, \pi)$$

In other words, the smaller the distance between prior and posterior in weight space, the better they will generalize also out of distribution. $\square$

## H    PROOF OF THEOREM 4.2

**Theorem 4.2.** *For any Lipschitz continuous loss $l : \mathcal{X} \to [0, 1]$ with Lipschitz constant $\lambda$, weighting function $w : \mathcal{X} \to [0, \beta]$ independent of the training set $Z = (x_i, y_i)_{i=1}^n$, and any density $p$, we have with probability at least $1 - \delta$ over the draw of $Z$:*

$$r_{exp}(f; u) \leq r_w(f; p_Z) + (\beta \lambda \, \mu + \|w\|_L) \, \mathbb{E}_{Z \sim p^n} [W_1(p, p_Z)] + 2\beta \sqrt{\frac{2 \ln(1/\delta)}{n}} + \delta(u, \hat{u}),$$

*where the classifier $f : \mathcal{X} \to \mathcal{Y}$ is a function dependent on the training data with Lipschitz constant at most $\mu$, $\delta(u, \hat{u}) = \int_x |u(x) - p(x)w(x)|dx$ is the $\ell_1$ distance between the uniform distribution $u$ and the re-weighted training distribution $\hat{u}(x) = p(x)w(x)$, $W_1(p, p_Z)$ is the 1-Wasserstein distance between $p$ and the empirical measure $p_Z$, and $\|w\|_L$ is the Lipschitz constant of $w$.*

*Proof.* The expected worst-case generalization error w.r.t. the uniform is given by

$$\sup_{f \in \mathcal{F}} (r_{\exp}(f; u) - r_w(f; p_Z)) = \sup_{f \in \mathcal{F}} (r_{\exp}(f; u) - r_w(f; p) + r_w(f; p) - r_w(f; p_Z))$$

$$\leq \sup_{f \in \mathcal{F}} (r_w(f; p) - r_w(f; p_Z)) + \sup_{f \in \mathcal{F}} (r_{\exp}(f; u) - r_w(f; p)).$$

Let us start with the first term:

$$\mathbb{E}_{Z \sim p^n} [T_1] = \mathbb{E}_{Z \sim p^n} \left[ \sup_{f \in \mathcal{F}} (r_w(f; p) - r_w(f; p_Z)) \right]$$

$$= \mathbb{E}_{Z \sim p^n} \left[ \sup_{f \in \mathcal{F}} \mathbb{E}_{z \sim p} [w(x) \, l(f(x), y)] - \mathbb{E}_{z \sim p_Z} [w(x)l(f(x), y)] \right]$$

For any $f \in \mathcal{F}$ and $z$, define $g(z) = l(f(x), y)$. By the sub-multiplicity of Lipschitz constants, we have

$$\|g\|_L \leq \|l\|_L \|f\|_L = \lambda \mu$$

Further,

$$\|w(x)g(x, y) - w(x')g(x', y')\|$$

$$\leq \|w(x)g(x, y) - w(x)g(x', y')\| + \|w(x)g(x', y') - w(x')g(x', y')\|$$

$$\leq \sup_x |w(x)| \|g(x, y) - g(x', y')\| + \sup_{x,y} |g(x, y)| \|w(x) - w(x')\|$$

$$\leq \sup_x |w(x)| \|g\|_L \|(x, y) - (x', y')\| + \sup_{x,y} |g(x, y)| \|w\|_L \|x - x'\|$$

Substituting $g(x, y) = l(f(x), y) \in [0, 1]$, we get

$$\frac{\|w(x)l(f(x), y) - w(x')l(f(x'), y')\|}{\|(x, y) - (x', y')\|} \leq \sup_x |w(x)| \|l\|_L \|f\|_L + \|w\|_L \frac{\|x - x'\|}{\|(x, y) - (x', y')\|}$$

$$= \beta \lambda \mu + \|w\|_L \frac{\|x - x'\|_2}{\sqrt{\|x - x'\|_2^2 + \|y - y'\|_2^2}}$$

$$\leq \beta \lambda \mu + \|w\|_L.$$

Next, denote by $h(z) = w(x)l(f(x), y)$ and let $\mathcal{H}$ be the corresponding hypothesis class. We have

$$T_1 = \mathbb{E}_{Z \sim p^n} \left[ \sup_{\|f\|_L \leq \mu} \mathbb{E}_{z \sim p} [w(x) \, l(f(x), y)] - \mathbb{E}_{z \sim p_Z} [w(x)l(f(x), y)] \right]$$

$$\leq \mathbb{E}_{Z \sim p^n} \left[ \sup_{h : \|h\|_L \leq \beta \lambda \mu + \|w\|_L} (\mathbb{E}_{z \sim p} [h(z)] - \mathbb{E}_{z \sim p_Z} [h(z)]) \right]$$

$$= (\beta \lambda \mu + \|w\|_L) \, \mathbb{E}_{Z \sim p^n} [W_1(p, p_Z)].$$

where in the last step, we used the Kantorovich-Rubenstein duality theorem. Moving on to the next term:

$$T_2 = \sup_{f \in \mathcal{F}} (r_{\exp}(f; u) - r_w(f; p))$$

$$= \sup_{f \in \mathcal{F}} \left( \int_x u(x) \, l(f(x), f^*(x)) \, dx - \int_x p(x)w(x) \, l(f(x), f^*(x)) \, dx \right)$$

$$= \sup_{f \in \mathcal{F}} \int_x (u(x) - p(x)w(x)) \, l(f(x), f^*(x)) \, dx$$

$$\leq \int_x |u(x) - p(x)w(x)|dx$$

$$= \delta(u, \hat{u})$$

where $\hat{u}(x) = p(x)w(x)$.

To characterize the generalization error difference between a random training set and the expectation we will need the following technical result:

**Lemma H.1.** *For any* $l : \mathcal{X} \to [0, 1]$*, the maximal generalization error*

$$\varphi_w(Z) = \sup_{f \in \mathcal{F}} (r_w(f, p) - r_w(f, p_Z)) \tag{11}$$

*obeys the bounded difference condition*

$$|\varphi_w(Z) - \varphi_w(Z')| \leq \frac{2}{n} \max_x |w(x)|, \tag{12}$$

*where we have used the notation*

$$Z = \{z_1, \ldots, z_j, \ldots, z_n\} \quad \text{and} \quad Z' = \{z_1, \ldots, z_j', \ldots, z_n\}. \tag{13}$$

*Proof.* We first rewrite the difference as

$$|\varphi_w(Z) - \varphi_w(Z')| = \left| \sup_{f \in \mathcal{F}} (r_w(f, p) - r_w(f, p_Z)) - \sup_{f \in \mathcal{F}} (r_w(f, p) - r_w(f, p_{Z'})) \right| \tag{14}$$

$$\leq \sup_{f \in \mathcal{F}} |r_w(f, p_Z) - r_w(f, p_{Z'})|, \tag{15}$$

where we have used that $\sup_x f_1(x) - \sup_x f_2(x) \leq \sup_x (f_1(x) - f_2(x))$ in the last step.

We continue by expanding the above expression:

$$\sup_{f \in \mathcal{F}} |r_w(f, p_Z) - r_w(f, p_{Z'})| = \sup_{f \in \mathcal{F}} \left| \frac{1}{n} \sum_{i=1}^n w(x_i) \, l(f(x_i), y_i) - \frac{1}{n} \sum_{i=1}^n w(x_i') \, l(f(x_i'), y_i') \right|$$

$$= \frac{1}{n} \sup_{f \in \mathcal{F}} \left| w(x_j) \, l(f(x_j), y_j) - w(x_j') \, l(f(x_j'), y_j') \right|$$

$$= \frac{1}{n} \sup_{f \in \mathcal{F}} \left| w(x_j) \, l(f(x_j), y_j) - w(x_j') \, l(f(x_j), y_j) + w(x_j') \, l(f(x_j), y_j) - w(x_j') \, l(f(x_j'), y_j') \right|$$

$$\leq \frac{1}{n} \sup_{f \in \mathcal{F}} \left( |l(f(x_j), y_j)| \, |w(x_j) - w(x_j')| + |w(x_j')| \, |l(f(x_j), y_j) - l(f(x_j'), y_j')| \right)$$

$$\leq \frac{2}{n} \max_x |w(x)|,$$

as claimed, where in the last step we used that the loss is between zero and one, whereas the weights are always positive. $\square$

Since $\varphi_w$ fulfils the bounded difference condition (see Lemma H.1), we can apply McDiarmid's inequality to obtain

$$\mathbb{P}\left[\varphi_w(Z) - \mathbb{E}_Z[\varphi_w(Z)] \geq \epsilon\right] \leq \exp\left(-\frac{\epsilon^2 n}{4 \max_x |w(x)|^2}\right) \tag{16}$$

This probability is below $\delta$ for $\epsilon^* \geq 2 \max_x |w(x)| \sqrt{\frac{\ln(1/\delta)}{n}}$, which immediately implies that

$$\mathbb{P}\left[\varphi_w(Z) < \mathbb{E}_Z[\varphi_w(Z)] + \epsilon^*\right] = 1 - \mathbb{P}\left[\varphi_w(Z) - \mathbb{E}_Z[\varphi_w(Z)] \geq \epsilon^*\right] \geq 1 - \delta,$$

from which we conclude that the following holds

$$\sup_{f \in \mathcal{F}} (r_w(f; p) - r_w(f; p_Z)) \leq \mathbb{E}_{Z \sim p^n}[\varphi_w(Z)] + 2 \max_x |w(x)| \sqrt{\frac{2 \ln(1/\delta)}{n}} \tag{17}$$

with probability at least $1 - \delta$. Combining (17) with the previous results yields:

$$\sup_{f \in \mathcal{F}} (r_{\exp}(f; u) - r_w(f; p_Z)) \leq \sup_{f \in \mathcal{F}} (r_w(f; p) - r_w(f; p_Z)) + \sup_{f \in \mathcal{F}} (r_{\exp}(f; u) - r_w(f; p))$$

$$\leq \mathbb{E}_{Z \sim p^n} [\varphi_w(Z)] + 2 \max_x |w(x)| \sqrt{\frac{2 \ln(1/\delta)}{n}} + \delta(u, \hat{u})$$

$$\leq (\beta \lambda \mu + \|w\|_L) \, \mathbb{E}_{Z \sim p^n} [W_1(p, p_Z)] + 2 \max_x |w(x)| \sqrt{\frac{2 \ln(1/\delta)}{n}} + \delta(u, \hat{u}),$$

as claimed. $\qquad\square$

# I   ESTIMATING THE UNIFORM EXPECTED RISK BY IMPORTANCE SAMPLING ON A VALIDATION SET

We consider that the validation samples used to estimate the uniform risk are drawn from some density $p(x)$ that is known up to normalization and are held out from training (crucially, $f$ is independent of the validation samples).

We will estimate $r_{\exp}(f; u)$ by the importance sampling estimate

$$r_{\hat{\rho}}(f; p_Z) = \frac{1}{n} \sum_{i=1}^{n} \hat{\rho}(x_i) \, l(f(x_i), y_i) \quad \text{with} \quad \hat{\rho}(x) = p(x)^{-1} \frac{n}{\sum_{x' \in Z} p(x')^{-1}} \propto \rho(x) = \frac{u(x)}{p(x)} \tag{18}$$

To apply the theorem of Chatterjee and Diaconis (Chatterjee & Diaconis, 2018), we note that for the 0-1 loss:

$$\|f\|_{L^2} = \sqrt{\mathbb{E}_{(x,y) \sim u}[l(f(x), y)^2]} = \sqrt{\mathbb{E}_{(x,y) \sim u}[l(f(x), y)]} = \sqrt{r_{\exp}(f; u)},$$

whereas the KL-divergence is given by

$$D_{\mathrm{KL}}(u\|p) = \sum_{x \in \mathcal{X}} u(x) \log \frac{1}{p(x)} - \sum_{x \in \mathcal{X}} u(x) \log \frac{1}{u(x)} = H(u, p) - H(u).$$

Suppose that $n = e^{D_{\mathrm{KL}}(u\|p) + t}$ for some $t > 0$ and fix

$$\varepsilon^2 = e^{-t/4} + 2\sqrt{\zeta}$$

with $\zeta \leq \mathbb{P}_{x \sim u} (\log(\rho(x)) < D_{\mathrm{KL}}(u\|p) + t/2)$. Then it follows from Chatterjee & Diaconis (2018) that

$$\mathbb{P}\left(|r_{\hat{\rho}}(f; p_Z) - r_{\exp}(f; u)| \geq \frac{2\sqrt{r_{\exp}(f; u)}}{1/\varepsilon - 1}\right) \leq 2\varepsilon.$$

