# OpenReview forum: "Generalizing to any diverse distribution: uniformity, gentle finetuning & rebalancing"
_ICLR.cc/2025/Conference — Submitted to ICLR 2025_

### Official Review · Reviewer_B5w2 · 2024-11-01

**Soundness:** 2
**Presentation:** 2
**Contribution:** 2
**Rating:** 5
**Confidence:** 3

**Summary:**

This paper proposes a new learning framework for studying the out-of-distribution generalization. They construct the uncertainty set by using the entropy of the data distribution. They define the worst-case risk in this uncertainty set by Distributionally diverse (DD) risk. They build a series of theorems containing: A classifier optimized for the uniform distribution will yield the smallest DD risk; the generalization bound of DD risk under the zero-one loss. They further provide two ways to improve the generalization ability, fine-tuning with penalty and rebalance through weighting. They verify the effectiveness of the proposed method on IwildCam, PovertyMAp, and ColorMNIST datasets.

**Strengths:**

The proposed theoretical framework is novel. The paper is well written.

**Weaknesses:**

The experimental results are not solid enough to make a conclusion. Although rebalancing (UMAP-8, label cond., WDL2) achieves good performance in "-90%" and therefore performs well in average score, it remains unclear whether weight distance to initialization is practically effective, as no results for this metric are provided on other datasets in Tables 1 and 2. Furthermore, Tables 1 and 2 exclude many relevant baseline methods—such as ARM, which performs well on ColoredMNIST, as well as VREx, RSC, and others.

Additionally, there is no universal criterion for determining which method is best suited to practical application. The authors apply different modeling approaches (e.g., WDL2 for ColorMNIST and UMAP for PovertyMap) depending on the dataset, yet provide no general guidance on how to select methods for other contexts.

**Questions:**

Is uniform distribution best in real applications? I believe image data are located in the low dimensional manifold. Why does this uniform distribution help generalization?

---

> ### Author Response · Authors · 2024-11-23
>
> Thank you for your review and for pointing out the novelty of our framework.
>
> **The experimental results are not solid enough to make a conclusion. Although rebalancing (UMAP-8, label cond., WDL2) achieves good performance in "-90%" and therefore performs well in average score, it remains unclear whether weight distance to initialization is practically effective, as no results for this metric are provided on other datasets in Tables 1 and 2. Furthermore, Tables 1 and 2 exclude many relevant baseline methods—such as ARM, which performs well on ColoredMNIST, as well as VREx, RSC, and others.**
>
> We believe that the experimental results provide a well rounded picture of the effect of rebalancing on worst-group accuracy. Even though the main contributions of this work like in the novel mathematical framework and learning theorerical results, we still consider 7 well known datasets and illustrate both the benefits and pitfalls of the proposed approach. The newly added Appendix C.3 also shows the effectiveness of rebalancing over many training runs and hyperparameter sets of the downstream classifier. There we also show how traditional validation-based model selection can lead to poor o.o.d. generalization and that weight distance based selection can help to select a more robust model.
>
> Regarding the additional baselines:
>
> - ARM performs worse than ERM on WILDS, according to the public leaderboard, whereas VREx, RSC are not reported. However, we run C-Mixup which is the state-of-the-art on the PovertyMap dataset. As seen in Table 2, C-Mixup performs on par with rebalancing. We also tried combining rebalancing with C-Mixup in Appendix C.4, with some success. We also ran C-Mixup modifed for categorical labels on iWildCam where it underperformed standard ERM.
> - Nevertheless, we stress that our objective is not to establish state-of-the-art results (we don't make any sota claim in the paper) but only to establish the empirical validity and relevance of our theoretical claims. To avoid missunderstandings, we also made this explicit in the updated manuscript stating "We compare to vanilla empirical risk minimization (ERM) and the baselines reported by the original studies; we do not claim state-of-the-art performance.” (1st paragraph of Section 5.2)
>
> **There is no universal criterion for determining which method is best suited to practical application. The authors apply different modeling approaches (e.g., WDL2 for ColorMNIST and UMAP for PovertyMap) depending on the dataset, yet provide no general guidance on how to select methods for other contexts.**
>
> The performance gains do depend on the quality of the density fit and, unfortunately, as we expose in Sec 5.3 and Appendix C.5, density estimation remains only partly solved.  We found that the use/type of dimensionality reduction and the conditioning on labels, often had a strong effect on the quality of density estimation and can thus influence downstream performance.
>
> Fortunately, there is a simple and principled way to determine the hyperparameters of the density estimation: use a validation set to and select the model that achieves the best negative-log-likelihood or, even better, the top downstream validation metric. Our protocol is explained in Appendix C.1. This is no different to how hyperparameter search is done across all methods.
>
> **Is uniform distribution best in real applications? I believe image data are located in the low dimensional manifold. Why does this uniform distribution help generalization?**
>
> This is a missunderstanding. We argue that the optimal distribution is the uniform distribution over the domain where the data are defined. In practice, this means that we should train on a uniform distribution on the manifold, not over the pixel space. Assigning probability mass to regions of the input space where no natural images live is, as one would expect, not meaningful. In practice, we approximate the data manifold by using a density estimator.

---

> > ### Comment · Reviewer_B5w2 · 2024-12-01
> >
> > Thank the authors for the response. I have carefully read the rebuttal. I understand the claim of the authors. However, I would suggest more experiments in the future version since it might be asked by other reviewers. I will keep the score.

---

> ### Author Response · Authors · 2024-12-03
>
> Dear reviewer B5w2, in accordance to your final remark, we have included results on the datasets suggested by DzLP in our latest general reply. As you can see from those results, the conclusions are in line with the experiments already present in the paper.

---

### Official Review · Reviewer_3tMa · 2024-11-02

**Soundness:** 2
**Presentation:** 4
**Contribution:** 2
**Rating:** 3
**Confidence:** 5

**Summary:**

This paper addresses risk minimization conditioned on covariates, i.e., worst-case risk as named in the paper, by introducing the learning objective of distributionally diverse risk that considers the worst-case risk over high-entropy distributions. The paper proceeds to consider reweighted risk on uniform distribution as a surrogate for the distributionally diverse risk, which leads to the classic importance weighting technique. Experiments are conducted on datasets with prevalent covariate shift.

**Strengths:**

This paper follows a clear and logical organization:
1. Worst-case risk should be considered for prediction under covariate shift.
2. The distribution diverse risk is a good surrogate for the worst-case risk.
3. The risk on uniform distribution is a good surrogate for the distribution diverse risk.

The first point is an important objective and recognized challenge for covariate shift generalization. The paper has also well addressed the third point by (1) showing reweighting data towards a uniform distribution yields the lowest distribution diverse risk within the framework of reweighted empirical risk minimization (Theorem 3.1), and (2) studying the generalization gap between distribution diverse risk and uniform distribution risk at both population and sample levels (Theorems 3.2 and 4.3).

The paper also does a good work in explicitly acknowledging possible limitations, which improves overall clarity. For example, the author notes that Theorem 3.1 is less applicable to the setting of supervised domain adaptation, which is just fine because learning under uncertainty of target distribution is actually more challenging. But the statement helps accurately contextualize the results in the literature.

**Weaknesses:**

I am not convinced by the paper that the distribution diverse risk is a good surrogate for the worst-case risk. The introduction has a short explanation that high-entropy test distributions as considered by the distribution diverse risk reflect natural distribution shifts, and high-entropy training and test distribution empirically yield good generalization in literature, which will benefit from more detailed elaboration. In the current submission, I find the argument is not supported, or even contradicted by the theory.

The distribution diverse risk (DD) converges to the worst-case risk until $\gamma \to \infty$ (eq 4), when DD risk reduces to the worst-case over **all** possible distributions X. This asymptotic behavior is satisfied by a large family of "uncertainty sets" of test distributions, e.g., when the radius of the uncertainty set of Distributionally Robust Optimization (DRO) tends to infinity, or when we redefine DD as distributions with a differential entropy near any real value by a gap $\gamma$. This asymptotic argument does not address why  DD is superior to the alternative designs of uncertainty sets, in terms of a surrogate for the worst-case risk, which is pivotal for the position of this paper in the broad literature of OOD generalization under covariate shift, e.g., DRO [1].

It is also revealed in Theorem 3.2 that the DD risk scales with $\Omega(\gamma)$ as $\gamma \to \infty$, indicating that the DD risk cannot be estimated by uniform distribution risk at all when we actually consider the objective of worst-case risk. The author also acknowledges that the theorem only applies to $\gamma = O(1)$, contradicting the setup for the worst-case risk. Therefore, I interpret from the paper that the DD risk is another heuristic design of the uncertainty set of target distribution, as similar in DRO. However, a stronger theoretical or empirical reasoning for this design is expected in order to consolidate this paper.

Minor points:

1. Implications for LLM and foundation models (L61-69) in the introduction can be over-claimed since they are not supported by theory and the LLM-free experiments. Moreover, importance weighting in general is ineffective for over-parameterized models [2]. The intuition is that over-parameterized models interpolate the samples with an almost zero loss on each sample, corresponding to $\epsilon = 0$ in Theorem 3.1. In this case, the sample risk does not uniformly bound the population risk because there are infinitely many interpolators on training data among which most are not generalizable. In this case, considering the maximum risk of predictors does not make sense. As is also stated by the author, Theorem 3.1 does not consider the inductive bias of optimizers, but generalization for over-parameterized models are only possible when implicit regularization is imposed by the learner.

2. 0-1 loss and the existence of a ground truth model $f^\star(x)$ is assumed throughout the theory. Combined together, they imply an impractically strong assumption that the classification task is noiseless, i.e. $f^\star(x)$ is binary rather than a probability, which is not even satisfied by the simple classic model of mixture of gaussians, which also appears as the first experiment.

3. Section 4.1 and 4.2 are less relevant to the sample-level analysis of the generalization behavior of DD risk. This is because the author characterizes the risk gap by the l1 distance between the empirical measure and the population measure. However, this distance $\delta(u,p)$ always equals 2 for any continuous measures u and discrete measures p, which makes eq. 8 ineffective when p is an empirical measure, the sum of n dirac measures. This happens to Theorem 4.1 as well. In fact, a standard PAC-Bayes bound will characterize the risk gap as the ratio between some divergence measure and a power of n. The absence of sample numbers in eq.8 and Theorem 4.1 cannot reflect the non-asymptotic behavior as $n \to \infty$.

4. ColoredMNIST is a dataset that contains large concept shift instead of covariate shift as claimed by the paper. This synthetic dataset flips the label of digits (y) with a probability that is correlated with the color (part of x, denoted by $x_2$). This correlation varies across splits, leading to concept shift ($y|x_2$ shift). If we denote the shape of the digits as $x_1$, the dataset features $y \not \perp x_2 | x_1$. This causes $y|x$ shift.

5. An exhaustive investigation of baselines is performed for ColoredMNIST. Reporting their results for iWildCam and PovertyMap can be more convincing. For example, the public benchmark of PovertyMap shows that the SOTA C-Mixup, some simple variant of Mixup in Table 3, is surprisingly effective (0.53 v.s. 0.47 as the best result in the paper).

[1] Duchi, J. C., Hashimoto, T., & Namkoong, H. (2019). Distributionally robust losses against mixture covariate shifts.

[2] Zhai, R., Dan, C., Kolter, J. Z., and Ravikumar, P. K. Understanding why generalized reweighting does not improve over ERM.

**Questions:**

1. In theorem 3.2, what is the choice of $\alpha$ that minimizes the upper bound?

---

> ### Author Response · Authors · 2024-11-23
>
> **I am not convinced by the paper that the distribution diverse risk is a good surrogate for the worst-case risk. The introduction has a short explanation that high-entropy test distributions as considered by the distribution diverse risk reflect natural distribution shifts, and high-entropy training and test distribution empirically yield good generalization in literature, which will benefit from more detailed elaboration. In the current submission, I find the argument is not supported, or even contradicted by the theory.**
>
> **The distribution diverse risk (DD) converges to the worst-case risk until $\gamma \rightarrow \infty$ (eq 4), when DD risk reduces to the worst-case over all possible distributions X. This asymptotic behavior is satisfied by a large family of "uncertainty sets" of test distributions, e.g., when the radius of the uncertainty set of Distributionally Robust Optimization (DRO) tends to infinity, or when we redefine DD as distributions with a differential entropy near any real value by a gap γ. This asymptotic argument does not address why DD is superior to the alternative designs of uncertainty sets, in terms of a surrogate for the worst-case risk, which is pivotal for the position of this paper in the broad literature of OOD generalization under covariate shift, e.g., DRO [1].**
>
> Thank you for your review. We humbly urge you to entertain the idea that we do have good answers for the points you have raised and to read with an open mind.
>
> We answer your main issue by bring forth 3 points that address a key misconception and we believe address your expressed concerns.
>
> **Major point 1:  we do not consider the objective of worst-case risk in this work**
>
> As we have stated in Section 2, trying to control the worst-case error is generally a lost cause in the context of learning. We instead put forth an alterative and milder objective (worst case diverse distribution — see below) that renders the problem tractable within some reasonable regime: as proven under constant gap /gamma = O(1), the DD risk in minimized by the uniform risk. Thus our theory supports (and does not contradict) our objective.
>
> **Major point 2: DD and DRO make very different assumptions about what types of distribution shifts can be expected.**
>
> You can find our answer in the main reply, as well as in the expanded the related work discussion in Appendix A (lines 820-828). We have also included Figure 5 which illustrates the conceptual differences between DD and DRO providing examples of test admissible distributions for DD that are inadmissible under DRO and vice versa.
>
> **Major point 3. Why the assumption of high test entropy (the assumption behind the DD risk formalism) is meaningful.**
>
> We have included a new section in Appendix B that justifies the assumption underlying the DD risk formulation from a mathematical, intuitive and empirical perspective. Please also see the corresponding section in the main reply.
>
> We hope that these help to convince you of the difference of our framework from DRO and provide further justification for our formalism.

---

> ### Author Response · Authors · 2024-11-23
>
> **Minor point 1a. Implications for LLM and foundation models (L61-69) in the introduction can be over-claimed since they are not supported by theory and the LLM-free experiments.**
>
> Noted. We have changed the language of the abstract to “our theory aligns with previous observations” rather than "provides a mathematical grounding for previous observations”. We also added the following phrase to the conclusion: “the observations we have made about how our theory aligns with previous evidence in the literature on the training of foundation models do not establish a causal relation between our theory and reality. Further work will be needed to rigorously establish these links.”
>
> **Minor point 1b. ... importance weighting in general is ineffective for over-parameterized models [2]. The intuition is that over-parameterized models interpolate the samples with an almost zero loss on each sample, corresponding to ϵ=0 in Theorem 3.1. In this case, the sample risk does not uniformly bound the population risk because there are infinitely many interpolators on training data among which most are not generalizable. In this case, considering the maximum risk of predictors does not make sense.**
>
> The empirical risk is considered in Thms 4.1 and 4.2 as deduced by the use of the empirical measure $p_Z$ on the RHS of the equation. The theorems show that, despite overparametrization, the generalization error can indeed be small if the distance $\delta(\pi, \pi_Z)$ between prior and posterior is small. Thus, the inductive bias of optimization and the learning algorithm more generally is implicitly captured in Theorems 4.1 and 4.2 through $\delta(\pi, \pi_Z)$.
>
> It's worth also commenting on the reference you quoted [3] which argues that after long training SGD with importance weighting (IW) converges to a similar solution as ERM:
>
> - The way IW is utilized in these works is different: though [3] references [4] to draw empirical evidence that IW doesn't work, [4] uses importance weighting to match a target distribution or to control for class imbalance and not to make the training data more uniform as we do here (hence the name rebalancing).
> - These works primarily focus on the effect of long training. [4] focuses on the inductive bias of SGD after long training rather than the benefit of IW (which is taken as granted). The paper shows that the tendency of SGD to ignore IW after long training can be compensated by early stopping (among other ways), which is standard in the OOD literature and we employ it here.
>
> [3] Understanding Why Generalized Reweighting Does Not Improve Over ERM. Runtian Zhai and Chen Dan and J Zico Kolter and Pradeep Kumar Ravikumar. ICLR 2023.
>
> [4] What is the Effect of Importance Weighting in Deep Learning?. Jonathon Byrd and Zachary C. Lipton. 2019.
>
>
>
> **Minor point 2. 0-1 loss and the existence of a ground truth model f⋆(x) is assumed throughout the theory. Combined together, they imply an impractically strong assumption that the classification task is noiseless, i.e. f⋆(x) is binary rather than a probability, which is not even satisfied by the simple classic model of mixture of gaussians, which also appears as the first experiment.**
>
> We argue that the assumption is not impractical as it reflects most tasks encountered in reality where real data come with binary labels and one needs to determine/minimize the classification error. It is also a standard model in learning theory. Note that the labels in the first experiment are discretised in agreement with our theory. Sorry this was not more clear—we will update the text accordingly.
>
> That being said, the results of Section 4 (Theorems 4.1 and 4.2 as well as equation 8) hold for a wider class of losses.

---

> ### Author Response · Authors · 2024-11-23
>
> **Minor point 3. Section 4.1 and 4.2 are less relevant to the sample-level analysis of the generalization behavior of DD risk. This is because the author characterizes the risk gap by the l1 distance between the empirical measure and the population measure. However, this distance $\delta(u, p)$ always equals 2 for any continuous measures u and discrete measures p, which makes eq. 8 ineffective when p is an empirical measure, the sum of n dirac measures. This happens to Theorem 4.1 as well. In fact, a standard PAC-Bayes bound will characterize the risk gap as the ratio between some divergence measure and a power of n. The absence of sample numbers in eq.8 and Theorem 4.1 cannot reflect the non-asymptotic behavior as $n \rightarrow \infty$.**
>
> We respectfully disagree. The standard PAC-Bayes bounds you mention focus on i.i.d. generalization, whereas the bounds in (8) and Theorem 4.1 concern o.o.d. generalization. Thus, even as the number of samples goes to infinity $n \to \infty$, the error will still remain non-zero when the training and test distribution are different. Indeed, we here consider the DD risk which focuses on the worst distribution of sufficient entropy (which is not the same as the training distribution).
>
> Note also that we don't expect (8) to be useful in most cases (as emphasized in the paper, lines 207-210).
>
> Theorem 4.2 on the other hand interpolates between an i.i.d. and o.o.d. bound and thus shows how the i.i.d. generalization converges to zero as the empirical measure converges to the expected one ($W_1(p, p_Z) \to 0$).
>
> **Minor point 4. ColoredMNIST is a dataset that contains large concept shift instead of covariate shift as claimed by the paper. This synthetic dataset flips the label of digits ($y$) with a probability that is correlated with the color (part of $x$, denoted by $x_2$). This correlation varies across splits, leading to concept shift ($y|x_2$ shift). If we denote the shape of the digits as $x_1$, the dataset features $y \not \perp x_2 | x_1$. This causes $y|x$ shift.**
>
> The reviewer gives a valid interpretation of ColoredMNIST as containing concept shift.
>
> However, similarly, one may interpret things from the perspective of covariate shift if they consider the different clusters forming the data (product between digits and colors) and then notice that the training and test distributions, though supported in all clusters, they concentrate their probability mass on different clusters. The true labeling function does not change, only the probability that each cluster is sampled.
>
> So, in this instance, the distribution shift can be both characterized as “covariate” and "concept” drifts.
>
> **Minor points 5. An exhaustive investigation of baselines is performed for ColoredMNIST. Reporting their results for iWildCam and PovertyMap can be more convincing. For example, the public benchmark of PovertyMap shows that the SOTA C-Mixup, some simple variant of Mixup in Table 3, is surprisingly effective (0.53 v.s. 0.47 as the best result in the paper).**
>
> Thank you for the bringing this to our attention.
>
> We have included C-Mixup in the revised version, also extending it to the setting with categorical labels so that we can evaluate in on iWildCam. Our best method performs similarly or better in all instances except in the o.o.d. sets of PovertyMap, where, as we explained in lines 466-467, the density fit is poor due to a noticeable domain shift (that our approach is not designed to account for). Note also that the reported numbers were obtained by rerunning the method so, as expected, small differences (within experimental variance) can be observed.
>
> We also stress that our objective is not to establish state-of-the-art results (we don't make any sota claim in the paper) but only to establish the empirical validity and relevance of our theoretical claims. To avoid missunderstandings, we also made this explicit in the updated manuscript stating "We compare to vanilla empirical risk minimization (ERM) and the baselines reported by the original studies; we do not claim state-of-the-art performance.” (1st paragraph of Section 5.2)

---

> ### Comment · Reviewer_3tMa · 2024-11-25
> **Reviewer's rebuttal response**
>
> Thank the authors for their thorough response, and my gratitude also goes to the other reviewers for their efforts. I appreciate the authors' feedback regarding the relation between DD risk and worst-case risk. The rebuttal has clarified that worst-case risk is not the learning objective of this paper. I suggest minor edits to the manuscript to emphasize this distinction and avoid misinterpretations. For example, “DD risk subsumes the worst-case risk as a special case” (L111) might give the impression that worst-case risk is covered with the learning objective of this work.
>
> My concern regarding enriched baselines (minor point 5) has been well addressed, and I appreciate the authors clarifying that achieving SOTA is not a claim of the paper.
>
> However, my major concern remains regarding the significance of the DD risk. While I fully agree that DD risk represents a distinct learning objective from DRO, I am not yet convinced that it addresses the challenges unresolved by DRO or other o.o.d. generalization techniques.
>
> "We know very little about the relation between the training and test distributions." It is a valid statement, and DRO is indeed only applicable to test distributions constrained to perturbations of the training distribution. However, the DD risk also imposes a constraint that the test distribution lies within a small distance of the uniform distribution. It would benefit the manuscript to provide further justification for why the uniform distribution is preferred over the training distribution as an anchor.
>
> "We wish to avoid judging the behavior of a classifier based on pathological examples, thus the test distribution should not assign high likelihood to any small set of examples". However, the uniform distribution also introduces the same issue, as it assigns disproportionately large weights to low-density regions in the original distribution. For example, if the data follows a sub-Gaussian distribution restricted to a compact support, reweighting to a uniform distribution could excessively focus the model on tail samples.
>
> "…test entropy has already been identified in the literature as an intuitive and empirically predictive measure for o.o.d. generalization". I agree that test entropy is a useful evaluation metric under distribution shifts, and entropy minimization is a well-established technique in test time adaptation. However, this paper seeks to go further beyond by using training distribution entropy as a criterion for sample reweighting. And it advocates for maximum entropy rather than minimum entropy. A more grounded discussion would strengthen the argument.
>
> In summary, I identify the “reference” distribution as the key difference between DRO and DD risk. A deeper and more rigorous discussion of the selection of the reference distribution would better motivate this work.

---

> > ### Author Response · Authors · 2024-11-25
> > **Reply to "Reviewer's rebuttal response"**
> >
> > Thank you for your prompt and kind reply. We are happy to read that we have addressed these points.
> >
> > **To summarize, your first new question pertains to the role of uniform as a “reference distribution” in the DD risk, and motivating why it is a good reference. You also argue that “It would benefit the manuscript to provide further justification for why the uniform distribution is preferred over the training distribution as an anchor.”**
> >
> > We emphasize that the special role of the uniform distribution (which leads to the interpretation as a reference) is *not* posed by assumption, but emerges implicitly as a consequence of the DD risk definition. This is proved in Theorem 3.1. So your question boils down to motivating the DD risk (and specifically our assumption of high test entropy). To that end, we have provided a mathematical, intuitive, and practical motivation of the DD risk definition in the "Justifying the DD risk formalism” section of the main reply.
> >
> > We have also updated the revised document (section Appendix B) to clarify the motivation of DD risk. Let us know if you would like us to also include some other point.
> >
> > **Your second question: “Why do we advocate for maximum entropy rather than minimum entropy. A more grounded discussion would strengthen the argument.”**
> >
> > The DD risk assumption has a maximum entropy principle flavor because we aim to impose as few assumptions as possible about the distribution. We stress that the connection between DD risk and max entropy principle is only partial: we place a lower bound bound on the entropy gap rather than maximizing it.
> >
> > Employing a minimum entropy assumption here would correspond to identifying the smallest set of examples on which the classifier is wrong — this is akin to a worst case analysis and admits the simple solution that every classifier learned from examples will generally have maximal error.
> >
> > Further motivation of the high test entropy assumption can be found in Appendix B, "Justifying the DD risk formalism", especially on subsections 1.2 and 2.
> >
> > **Your last question: “If the data follows a sub-Gaussian distribution restricted to a compact support, reweighting to a uniform distribution could excessively focus the model on tail samples.”**
> >
> > You make a great point: if the training distribution is subgaussian there might be little evidence about the behavior of the true labels in regions of low probability. Thus, we are likely to make mistakes if we over-emphasize these regions.
> >
> > That's exactly what is shown in Theorem 4.2, which takes into account the total error when learning from examples. (Theorem 3.1 does not account for the i.i.d. generalization error.) As can be seen, the theorem studies arbitrary weighting functions (rather than only importance weights) and exposes a trade-off between approximating the uniform distribution as a proxy for o.o.d. generalizaiton (last term) and minimizing the i.i.d. error (first and second term on the RHS). The reviewer can confirm that, in the case of importance weights with a subgaussian distribution, $\beta$ would be very large leading to a large error.
> >
> > To account for this, in lines 297-303 of the manuscript we explain how clipping the importance weights is important to control the total error. The use of clipping allows us to focus on the region of the domain where the training distribution contains large probability mass. We evaluated the effect of clipping in Section 5.1 and also utilized it in all our experiments.
> >
> > **Overall.** We will be happy to discuss further till you feel your concerns are addressed. We also hope to see you update your score in a way that reflects your updated beliefs about the work.

---

> > > ### Comment · Reviewer_3tMa · 2024-12-03
> > >
> > > Thank you to the authors for revisiting the major points in the revised manuscript. I note that most of these points were already reflected in my previous response. I appreciate the clarifications provided. However, I would like to maintain my rating because of my remaining concern with respect to the significance of the DD risk.

---

### Official Review · Reviewer_6w8L · 2024-11-03

**Soundness:** 3
**Presentation:** 2
**Contribution:** 2
**Rating:** 5
**Confidence:** 3

**Summary:**

The paper aims to address the challenge of out-of-distribution (o.o.d.) generalization in machine learning by proposing a framework centered around the concept of distributionally diverse (DD) risk, which aims to minimize the worst-case error across all sufficiently diverse test distributions within a known domain. The paper presents theoretical findings and practical remedies for training models that generalize well under any diverse test distribution. Key contributions include proving the optimality of training on a uniform distribution, analyzing the impact of non-uniformity through finetuning and rebalancing, and providing empirical evidence supporting the theory across various tasks involving distribution shifts.

**Strengths:**

1. The paper introduces a fresh perspective on OOD generalization by focusing on the DD risk, which is a significant departure from traditional domain adaptation and generalization approaches.

2. The theoretical analysis provides mathematical grounding for the optimality of training on uniform distribution and the impact of non-uniformity on generalization.

3. The paper offers practical solutions such as gentle finetuning and rebalancing to mitigate the effects of non-uniformity when uniform samples are unavailable, which are valuable for real-world applications.

4. With the development of basic models and large-scale models, it becomes increasingly important to maintain robustness when the distribution of training and test data is very different. The emphasis on uniform training in this paper provides new ideas for large model training. At the same time, fine-tuning and rebalancing strategies also have application potential in the actual training of basic models, providing an important reference for future basic model design and training optimization.

**Weaknesses:**

1. Given the problem formulation in this paper, the solution seems rather straightforward. The stated objective is to control "the worst-case error across all distributions with sufficiently high entropy." The approach borrows from TTA by adjusting the training distribution to control Distributional Discrepancy (DD). However, the paper does not adequately justify why DD is a truly reasonable metric and what practical physical interpretations it has in real-world applications.
2. The practical effectiveness of the proposed method is questionable. For example, in Tables 2 and 3, rebalancing does not show a significant improvement over ERM. The final performance gains seem to stem from various tricks (e.g., UMAP, label conditioning, etc.), which are inconsistently applied across different experiments. This suggests to me that the results might be due to data tuning rather than a genuinely effective method applicable to diverse scenarios.
3. The quantitative analysis of different strategies is not in-depth enough. For example, there is no detailed data to support the applicability and effect of the rebalancing strategy under different types of distribution shifts. In addition, there is a lack of systematic analysis of the sensitivity of fine-tuning parameters (such as fine-tuning step size, number of iterations, etc.) and their effects.
4. The paper's presentation could benefit from several improvements. For instance, Table 2 references "UMAP" and Figure 3 mentions "WDL2" without providing explanations in the corresponding captions or within the main text. This lack of clarification might hinder reader comprehension.

**Questions:**

Please see weaknesses.

---

> ### Author Response · Authors · 2024-11-23
>
> Thank you for your review and appreciating our fresh perspective. We adress the weekness you have outlined in turn.
>
> **1. Given the problem formulation in this paper, the solution seems rather straightforward. The stated objective is to control "the worst-case error across all distributions with sufficiently high entropy." The approach borrows from TTA by adjusting the training distribution to control Distributional Discrepancy (DD). However, the paper does not adequately justify why DD is a truly reasonable metric and what practical physical interpretations it has in real-world applications.**
>
> The main contribution of this work entails formulating and motivating the *distributional diversity* (DD) objective, and deriving rigorous theory that connects it with simple and practical approaches for mitigating the effect of distribution shift.
>
> To address your comment about adequate justification,  we have expanded our justification of the DD risk in the main reply and in Appendix B of the updated manuscript. Therein, we make a case that DD is mathematically, conceptually, and empirically/practically a reasonable formalism.
>
> We agree that the rebalancing solution is straightforward given the previous theory developed here but is not obvious otherwise. Though importance weights (IW) are a fundamental concept and have been used extensively for variuos purposes (e.g., in estimation, TTA), our analysis justifies a new use: using IW to rebalance the training data towards the uniform distribution. Our work provides rigorous and non-trivial analysis of the benefits of this new application of IW.
>
> We hope the above address your remark. We are happy to include further changes in the paper to address any remaining concerns.
>
> **2. The practical effectiveness of the proposed method is questionable. For example, in Tables 2 and 3, rebalancing does not show a significant improvement over ERM. The final performance gains seem to stem from various tricks (e.g., UMAP, label conditioning, etc.), which are inconsistently applied across different experiments. This suggests to me that the results might be due to data tuning rather than a genuinely effective method applicable to diverse scenarios.**
>
> The performance gains do depend on how good the training density is fit an as we discuss in Sec 5.3 and Appendix C.5, density estimation remains only partly solved.  Indeed, we found that the use/type of dimensionality reduction and the conditioning on labels, often had a strong effect on the quality of density estimation and can thus influence downstream performance.
>
> To determine which embedding projection method is used before fitting the density estimation, we perform a small grid search judged by final validation performance, as described in Appendix B.1. Please note, that UMAP and label conditioning that you mention are only used for fitting the density estimator and thus getting better sample weights and not for training the actual classification model. The classification model training and all the hyperparameters exactly match that of ERM.
>
> Since for iWildCam and ColorMNIST datasets label-reweighting was tested in the orginal studies (e.g., “Label reweighted” ERM model in Table 1), we include a version where we fit a label conditioned density estimator. Fitting a label conditioned density estimator is direct adaptation of this in our framework. We do not apply label weighting in PovertyMap dataset (Table 2) as it is a regression and not a classification task.
>
> With a tuned density estimator, we observe an improvement in worst group pearson correlation of 0.58→0.66 & 0.57→0.65 (in distribution set) 0.51→0.53 & 0.44→0.47 (o.o.d set) in Table 2 and worst-group binary accuracy of 0.10→0.37 in Table 3. Given how challenging these benchmarks are known to be and the performance of the baselines, we see this as a noteworthy improvement. The smallest of these improvements corresponds to the o.o.d. test set of the PovertyMap which, as seen in Fig 4b, is assigned very low likelihood by the density estimator indicating a domain shift (our theory doesn't claim to help in this case).  Please note, that for ColorMNIST (Table 3), we show in Figure 9, that o.o.d. performance consistently improves over a wide range of different hyperparameters, indicating that the provided results are robust.
>
> We should also mention that our results suggest that future progress in density estimation can unlock additional benefits.

---

> ### Author Response · Authors · 2024-11-23
>
> **3.1 There is no detailed data to support the applicability and effect of the rebalancing strategy under different types of distribution shifts:**
>
> Our evaluation focuses on tasks with "covariate shift” because this is what our theory motivates. As explained in the paper (line 422), we don't aim to address other types of distribution shifts which go beyond our theoretical analysis. Nevertheless, we considered a total of 7 challenging datasets from well known benchmarks in Section 5 and Appendix C.5 so our results illustrate the benefits and pitfalls on a wide range of scenarios.
>
> We have amended Section 2 (lines 107-109) to emphasize that our paper focuses only on covariate shift and does not address other types of shifts which remain hard open problems.
>
> **3.2. there is a lack of systematic analysis of the sensitivity of fine-tuning parameters (such as fine-tuning step size, number of iterations, etc.) and their effects.**
>
> As explained in the Appendix C.1, we follow the public benchmarks and evaluate all methods using the default hyperparameters recommended by the method authors (those exactly matching the hyperparametres of ERM in respective tasks). Tuning the hyperparameters could provide further boost for our approach but we avoided doing so because our objective is not to show sota results but to exemplify how our theory-derived insights can be practically advantageous.
>
> Nevertheless, following your suggestion, in Figure 9 (Appendix C.3) we showcase how the rebalancing affects the generalization performance over multiple training runs (20 different hyperparameter sets and three random trials). There we see that rebalancing quite consistently improves generalization.
>
> **4. The paper's presentation could benefit from several improvements. For instance, Table 2 references "UMAP" and Figure 3 mentions "WDL2" without providing explanations in the corresponding captions or within the main text. This lack of clarification might hinder reader comprehension.**
>
> Thank you for spoting this. UMAP was indeed not mentioned and we fixed this.

---

> > ### Comment · Reviewer_6w8L · 2024-11-30
> >
> > I thank the authors for the response. While the authors have provided additional descriptions, I am still not totally convinced that DD is a reasonable metric with significant practical implications. I believe the arguments made from the mathematical perspective and intuitive motivation do not directly lead to DD, and other similar ideas like groupDRO could also be applicable.
> >
> > Furthermore, I think the reason why TTA algorithms use entropy as an optimization metric is due to the lack of labels for test samples and the need for real-time adaptation. However, optimizing entropy does not perfectly align with the actual accuracy performance of the model. Many TTA methods need to incorporate additional constraints to prevent model performance from collapsing.
> >
> > Therefore, if the authors could provide further arguments to demonstrate that DD is superior to existing DRO-like metrics in terms of the mathematical perspective and intuitive motivation mentioned in the response, I believe it would significantly increase the persuasiveness of their claims.

---

> > > ### Author Response · Authors · 2024-11-30
> > > **Additional motivation wrt (group) DRO & why arguments about entropy in TTA don't apply**
> > >
> > > Thank you for your comment.
> > >
> > > We provide the requested motivation w.r.t. (group) DRO and TTA below. If further explanations are required, we will be happy to provide them.
> > >
> > > **1. Additional motivation of DD w.r.t. (group) DRO:**
> > > - **DRO makes the strong assumption that the test distribution will be close to the training distribution, whereas DD does not make such an assumption**. It is known in the literature (see references in main reply) that the “small train-test diversgence” assumption does not hold in many cases. For examples of test distributions that are admissible by DD and not DRO, please check Figure 5.
> > > - **group DRO assumes knowledge of group information, whereas DD doesn't**:  “We assume that we know which group each training point comes from—i.e., the training data comprises (x, y, g) triplets" [1]. Thus, group DRO benefits from proviledged information and it’s not entirely fair to compare it with our approach.
> > > - **empirically DD significantly outperformed group DRO**, with worst-case group improvements of 20%, 49%, and 366% on the PovertyMap,  iWildCAM, and ColorMNIST datasets respectively. These large improvements provide empirical evidence that the assumption made by DD (high test entropy) aligns better with practice.
> > >
> > > **2. Why arguments about entropy in TTA (e.g., [2]) don't apply to DD.**
> > >
> > > Recall that TTA assumes that the unlabeled test data are known, whereas in our framework this information is unavailable.
> > >
> > > Hence the two frameworks measure and use entropy in different ways:
> > >
> > > - **TTA considers the predictive entropy H( \hat{y} | x )**, where \hat{y} is the prediction of the trained model on the unlabeled test data x;
> > > - **DD considers the unlabeled data entropy H(x)**
> > >
> > > Thus, the entropies studied in the two frameworks are different and we cannot draw relations between them.
> > >
> > > [1] DISTRIBUTIONALLY ROBUST NEURAL NETWORKS FOR GROUP SHIFTS: ON THE IMPORTANCE OF REGULARIZATION FOR WORST-CASE GENERALIZATION. Shiori Sagawa, Pang Wei Koh, Tatsunori B. Hashimoto, Percy Liang. ICLR 2020
> > >
> > > [2]  Entropy is not Enough for Test-Time Adaptation: From the Perspective of Disentangled Factors. Jonghyun Lee, Dahuin Jung, Saehyung Lee, Junsung Park, Juhyeon Shin, Uiwon Hwang, Sungroh Yoon. 2024

---

### Official Review · Reviewer_DzLP · 2024-11-03

**Soundness:** 3
**Presentation:** 3
**Contribution:** 3
**Rating:** 5
**Confidence:** 5

**Summary:**

This paper proposes the distributionally diverse risk to address the out-of-distribution generalization problem. The DD risk takes the similar formulation as the DRO risk with the uncertainty set captured by entropy. Based on the DD risk, the authors demonstrate the optimality of the uniform distribution. Experimental results validate the effectiveness of the proposed reweighting/rebalancing methods.

**Strengths:**

- The authors propose the distributionally diverse risk and derive risk bounds for the DD risk.
- The authors demonstrate the optimality of the uniform distribution under their uncertainty set characterized by the entropy.
- Various experimental results are given to support their methods.

**Weaknesses:**

Given the theoretical and experimental results, my major concerns include:
- Relationship with Distributionally Robust Optimization: The proposed DD risk has similar formulation as the DRO risk, and the only difference is that the authors use entropy to capture the difference between distributions. In DRO literature, there are works using $f$-divergence, Wasserstein distance, MMD distance, etc. For different distance metrics used in DRO, there are also works to derive the risk bounds (for example, see [1,2]). What is the advantage of this specific kind of distance metric? Why can it lead to better generalization performance compared with others? I would suggest the authors providing a more comprehensive illustration on this.
- As the authors demonstrate, the optimality depends on the "entropy" selected for the uncertainty set. And the result is more like: the uniform distribution is optimal if we consider this specific kind of DRO risk (or the DD risk). Can the authors demonstrate why this specific formulation is better?
- As for the experiments on real-world datasets, can the authors demonstrate in detail how the density estimator is learned? (I read the appendix but found it is not so clear.) Are the embeddings coming from a pre-trained ResNet-50? How is the effect of the density estimator on other datasets? I think the density estimation itself may be as hard as the original prediction task. How can the authors guarantee that the learned estimator is good? And when it is bad, will it even hurt the model performance?
- What if there are noisy data? In the DRO literature, there is a phenomenon named "over-pessimism", i.e. the worst-case risk may be too unrealistic and thus hurt the model learning. I do think it will hold in the formulation proposed in this work. For example, when doing reweighting or rebalancing, what if some noisy data or outliers are given a much higher sample weight?



[1] Variance-based regularization with convex objectives. John Duchi, Hongseok Namkoong.
[2] Learning models with uniform performance via distributionally robust optimization. John Duchi, Hongseok Namkoong.

**Questions:**

Please refer to the weaknesses.

---

> ### Author Response · Authors · 2024-11-23
>
> Thank you for your review. We answer your concerns below.
>
> **Relationship with Distributionally Robust Optimization: The proposed DD risk has similar formulation as the DRO risk, and the only difference is that the authors use entropy to capture the difference between distributions. In DRO literature, there are works using divergence, Wasserstein distance, MMD distance, etc. For different distance metrics used in DRO, there are also works to derive the risk bounds (for example, see [1,2]). What is the advantage of this specific kind of distance metric? Why can it lead to better generalization performance compared with others? I would suggest the authors providing a more comprehensive illustration on this.**
>
> Thank you for bringing forth this important point and giving us the chance to explain how the DD risk framework is different from DRO.
>
> You can find our answer in the main reply, as well as in the expanded the related work discussion in Appendix A (lines 820-828). We have also included Figure 5 which illustrates the conceptual differences between DD and DRO providing examples of test admissible distributions for DD that are inadmissible under DRO and vice versa.
>
> **As the authors demonstrate, the optimality depends on the "entropy" selected for the uncertainty set. And the result is more like: the uniform distribution is optimal if we consider this specific kind of DRO risk (or the DD risk). Can the authors demonstrate why this specific formulation is better?**
>
> First, as explained in the main reply, the DD formulation is different from DRO because it changes the paradigm away from "the test distribution contains a bounded shift from the training distribution” to “the test distribution can be any arbitrary diverse distribution in the same domain". The DD risk formulation thus carries the benefit of not making strong assumptions about the training and test distributions being close — this assumption is often invalid in the real world (see main reply for references).
>
> Second, the superior numerical results provide further evidence of the usefulness of our formulation.
>
> We also argue that, it is beneficial to the community to investigate different perspectives to the important problem of o.o.d. generalization. Indeed, the DD and DRO frameworks make different assumptions and end up with different solutions. For instance, our results show that training on the uniform distribution is optimal for any (non pathological) entropy gap \gamma! This is in stark contrast to what one would obtain in DRO where as the test-train distance becomes smaller, DRO becomes more similar to ERM.

---

> ### Author Response · Authors · 2024-11-23
>
> **As for the experiments on real-world datasets, can the authors demonstrate in detail how the density estimator is learned? (I read the appendix but found it is not so clear.) Are the embeddings coming from a pre-trained ResNet-50?**
>
> We are happy to provide further details on the density estimators. The updated Appendix C.1 contains in detail the protocol we used to identify the hyperparameters (grid search with a validation set). The general pipeline is to take a pre-trained model, that acts as a featurizer, extract embeddings for the whole training set. Optionally, the embeddings are projected to a lower dimensional space and normalized, to make the density estimation task easier. This is the part on which we perform a grid search, considering PCA projection, UMAP projection or no projection, together with normalizing the vectors by their maximum lenght, or by standard deviation of each dimesion over the whole training set or not normalizing them. Then we train a masked autoregressive flow (MAF) to fit the density function on the training set embeddings  (random 10% is left out for validation). The 10% held out set is used for early stoping of MAF training. The MAF hyperparameters are fixed for all real-world tasks(2-layer MLP per MAF layer, hidden dimension of 256, 10 MAF layers). These correspond to common values found in the literature, that in preliminary experiments we found to be quite robust across tasks.
>
> In the original WILDS benchmarks a featurizer model is used with a classification/regression head added to it.  The featurizer and the head are finetuned on the given task. The featurizer models used in the WILDS benchmarks differ for different datasets, but are generally pre-trained (this is detailed in the WILDS paper). To fit the densities we simply use the featurizer model used in WILDS for the given task to produce the embeddings on which the density is fit. For ColorMNIST, we used a ResNet-50 pre-trained on ImageNet, as the original ColorMNIST experimental setup trains a custom CNN from scratch.
>
> **How is the effect of the density estimator on other datasets? I think the density estimation itself may be as hard as the original prediction task. How can the authors guarantee that the learned estimator is good? And when it is bad, will it even hurt the model performance?**
>
> Density estimation can be indeed hard for some datasets though it is unclear to us if it's as hard as the original task. A major benefit of density estimation is that, whereas the prediction task is supervised, the density estimation does not require labels and can thus benefit from additional data.
>
> A simple way to ensure that the density estimator is good is by checking the validation negative log-likelihood is small and by visualizing the likelihood distributions for the training and test set as we did in Figures 4 and 9. Telltale signs that the density fit is bad include: likelihood below zero for the training distribution (undefitting) or small likelihood for the validation set (overfitting), or both (underfitting). In our experiments we tuned the projectiona and normalization used for the embeddings on which the density estimator is fit as explained in Appendix C.1, choosing the setup with the best validation loss (for WILDS datasets, that is the o.o.d. validation loss, by the standard benchmark setup).
>
> In Tables 6-9 we report results for datasets with a poor density fit. Thankfully, even in this case the effect of rebalancing is generally not detrimental leading to performance similar to ERM (up to experimental variance).
>
> **What if there are noisy data? In the DRO literature, there is a phenomenon named "over-pessimism", i.e. the worst-case risk may be too unrealistic and thus hurt the model learning. I do think it will hold in the formulation proposed in this work. For example, when doing reweighting or rebalancing, what if some noisy data or outliers are given a much higher sample weight?**
>
> Excellent point. Our analysis in Section 4.3 provides a similar insight: the test error bound of Theorem 4.2 can be broken to two parts, with the rightmost term (that captured OOD error) benefiting from rebalancing, whereas the left-most term (capturing IID error) being affected negatively through $\beta$ by up-weighting outliers. This finding motivated us to clip the importance weights in the mixture of gaussians experiment.  This is fully aligned with our Theorem 4.2 which bounds the test error for any arbitrary weighting function and not only for importance weights.
>
> We enhanced our explanation (last sentence of Section 4.3) to make the relation between $\beta$, clipping and outliers explicit.

---

> > ### Comment · Reviewer_DzLP · 2024-11-25
> >
> > Thank you for your rebuttal. However, I am still concerned about the rationales of the DD risk (the uniform distribution), the incorporation of a model to estimate $p(X)$, and also the experimental results (where the improvements are not large, and the authors did not conduct experiments on some typical datasets like Waterbirds and MultiNLI). Also, there are some very simple reweighting methods with really nice results [1], which makes me doubt the meaning of the proposed complicated method.
> >
> > Therefore, I would like to maintain my score.
> >
> > [1] Simple data balancing achieves competitive worst-group-accuracy. CLeaR, 2022.

---

> ### Author Response · Authors · 2024-11-25
> **reply to "Official Comment by Reviewer DzLP"**
>
> “I maintain doubts about xx” to better understand your concerns: can you explicitly list your concerns that have not been addressed?
>
> **Wrt the proposed simple baseline: “Very simple reweighting methods with really nice results [1]”**
>
> **TLDR:** we already compare to the method and it does not perform strongly.
>
> The work you referenced evaluates reweighting based on either **group labels** or **target labels**, whereas we rebalance based on inputs.
>
> - **Group reweighting performs the best because it uses additional information.** Having group labels is a strong assumption that renders this method inappropriate for our setup. Yet, intuitively, group reweighting often ends up making the data more uniform in the input space, which is in alignment with our theory.
> - **Label reweighting did not perform strongly on the original paper or our experiments.** We evaluated this method on iWildCam (method “Label reweighed”) and it can be seen that all variants of our approach provide consistent improvement over it and the best variant achieves a large worst-group improvement: test i.d. accuracy of 77.5% (label reweighing does 70.8%) and test o.o.d. accuracy of 77.4% (label reweighing does 68.8%). ColorMNIST is already label balanced, so label reweighting is equivalent to ERM. PovertyMap is a regression task and thus there are no discrete labels.
>
> We will include the reference in the updated manuscript.
>
> **Addressing doubts about “the rationales of the DD risk (the uniform distribution)”:**
>
> - Justification: We have laid out a motivation clearly, stating how the uniform distribution optimality provably emerges from an entropy constrain and that the latter (akin to the well known max entropy principle) ensures that we make as few assumptions as possible while also not placing large probability mass on any small set of events.
> - We have further provided an intuitive and empirical motivation of our choice of entropy as a measure.
> - Our theory leads to practical algorithms that lead to quantifiable benefits over methods that make alternative assumptions
>
> What do you find false about our arguments? If you make your concerns more explicit it will help us to address them.
>
> **Addressing doubts on the empirical performance and evaluation**:
>
> - **Additional datasets.** We focused on the established WILDS and DomainBed benchmarks and have selected 7 representative datasets, which is a larger number of datasets that even many empirical works include. If you would have brought up the Waterbirds and MultiNLI datasets earlier in the review process we would have been happy to include experiments on these. We can commit to run experiments also on these datasets, but likely the results won't be ready in time for the discussion.
> - **Simplicity.** We argue that the proposed approach or rebalancing is conceptually very simple and, is already employed in a primitive form (based on clustering) by foundation models such as AlphaFold and ESM. The only complicated thing is density estimation, but this is a fundamental problem for which we have already some good algorithms and many people are working to improve it further.
> - **Empirical performance.** As can be deduced by the performance of baselines and the public leaderboards, these datasets are very challenging. So the improvements shown (in terms of worst-group accuracy) are non-trivial. We are not aware of any other method that achieves better results consistently on these three datasets. Yet, we restate that our purpose is not to establish state-of-the-art results but to corroborate our theory empirically.
>
> To ground the discussion on empirical evidence, if you find the achieved performance not convincing, can you provide us a with a reference to a method that consistently achieves better results on these datasets without additional information (such as knowledge of group labels)?

---

> ### Author Response · Authors · 2024-12-03
>
> Dear reviewer DzLP, see our latest general comment, for the results on Waterbirds and MultiNLI datasets you asked about.
> As you can see the conclusions follow the ones already present in the paper.

---

### Author Response · Authors · 2024-11-23
**General reply cont.**

### 2. Explaining how DD risk and DRO differ

**TLDR**: The DD risk and DRO approaches make starkly different assumptions about the relation between the training and test distribution and lead to different theoretical arguments and practical solutions.

As a reminder, the “distributionally robust stochastic optimization (DRO) framework (..) learns a model providing good performance against perturbations to the data-generating distribution" [3], whereas we here advocate for learning a model that exhibits good performance against any high entropy distribution in the same domain.

**Similarities.** Both approaches consider the worst case behaviour over a set of distributions. Further, at the limit, both DRO and DD converge to the worst case risk over the domain.

**Differences.** The assumptions posed by the two frameworks about the relation of the training and test distribution are starkly different:

- DRO assumes that the test distribution is close to the training distribution. The small train-test discrepancy assumption, though simple & intuitive is not always valid. To account for actual distribution shifts encountered in the wild, often very wide uncertainty sets are needed around the training distribution. Further, as also mentioned in the justification of DD risk, train-test discrepancy has been empirically found to correlate poorly with generalization [1] and concurrent theoretical work also advocates against it [2]
- We don't consider the worst-case bounded distribution shift (as is done in DRO) but the worst case risk under any distribution of sufficient entropy. As such, the training distribution holds no special role in our framework, whereas in practice DRO defines the “center of mass” of the set of admissible test distributions.

The above are consequential conceptual and practical differences:

- Conceptually, whereas in DRO one needs to think about how similar is the test distribution to the training distribution, we here consider arbitrary distributions on the test domain and pose a constraint on entropy. This is a change in paradigm.
- Practically, the two approaches lead to very different solutions to the problem and, as we show empirically, our approach can yield quantifiable benefits.

We have significantly expanded the related work discussion in Appendix A (lines 820-828) to highlight these differences and to include the references kindly provided by the reviewers. We have also included Figure 5 which illustrates the conceptual differences between DD and DRO by providing examples of admissible test distributions for DD that are inadmissible under DRO and vice-versa.

[1]  An empirical investigation of
domain generalization with empirical risk minimizers. Ramakrishna Vedantam, David Lopez-Paz, and David J Schwab. NeurIPS 2021.

[2] Beyond Discrepancy: A Closer Look at the Theory of Distribution Shift. Robi Bhattacharjee, Nick Rittler, Kamalika Chaudhuri. ArXiv *29 May 2024.*

[3] Learning Models with Uniform Performance via Distributionally Robust Optimization. John Duchi, Hongseok Namkoong. 2018

---

### Author Response · Authors · 2024-11-23
**General reply**

We thank all reviewers for their work and comments.

The reviews have acknowledged the novelty of our perspective within the imporant problem of o.o.d. generalization, found the paper well-written, commented positively on the practical applicability of our theory, and have not identified any errors in our theorems.

There were two especially important questions that we received:

1. to provide further justification of our "distributionally diverse (DD) risk” formalism
2. to clarify the relation of our approach with distributionally robust optimization (DRO)

We provide concrete answers to these two questions below. All other questions are answered at individual replies. We have also updated the manuscript with more detailed discussion of these main points as well as with clarifications and new empirical results requested by the reviewers.

We hope that the reviewers read our answers with open mind, giving our work a fair and equitable treatment.

### 1. Justifying the DD risk formalism

The DD risk entails measuring the behavior of a classifier across any test distribution of sufficiently high-entropy within a domain. In the following, we justify the definition from a mathematical, intuitive, and empirical perspective.

1. **Mathematical perspective.** Our definition of DD risk follows from two *desiderata* on how to model the generalization of a predictive model under covariate shift:
    1. **We can assume to know very little about the relation between the training and test distributions.** Unfortunately, the distance between training and test distribution (also referred to as train-test discrepancy) is often large in practice, discouraging stronger assumptions. This is supported by the literature where, train-test discrepancy has been empirically found to correlate poorly with generalization [1]. Concurrent/independent theoretical work also advocates against it [2].
    2. **Since we wish to avoid judging the behavior of a classifier based on pathological / overly specific examples, we care most about distributions that do not assign overly high likelihood to any small set of examples.** For context, desideratum 2 directly addresses the reason why worst-case analysis is non-meaningful in learning: if no such constraint is placed, an adversary may always construct a pathological test distribution that concentrates all its mass on a small set of inputs where the predictive model is wrong (in general, such a set can always exist when learning from examples). We directly avoid this situation by asserting that no such small high likelihood set can exist.

    It can be easily confirmed that asserting that the test distribution entropy is high satisfies both these desiderata. Lower bounding test entropy ensures that the spread of a distribution is large, without imposing any further assumptions about the relationship between train and test distributions.

2. **Intuitive motivation.** A generally performant model is one that generally performs well on the many test instances that the world throws at it—even if it might fail in specific instances. In other words, though we cannot hope that our model will generalize to any arbitrary sharp test distribution, we may expect that it will perform well on test distributions that are more spread out and thus more typical of the data domain.

    Our assumption of high test entropy exactly corresponds to the minimal and intuitive assumption that the classifier will be evaluated on a diverse set of possible inputs, rather than on few pathological examples.

3. **Empirical justification.** Finally, to emphasize the practical applicability of our assumption in real-world applications, we remark that, as mentioned in the introduction, test entropy was identified [1] as the most predictive measure for o.o.d. generalization in a recent thorough empirical study, significantly outperforming many other intuitive measures across multiple datasets.

These points justify our assumption as grounded in previous work, minimal, intuitive, and practically applicable.

Aiming to make these points more easily accessible to future readers, we have included a relevant new section in Appendix B of the revised document.

---

### Author Response · Authors · 2024-11-27
**Further evidence supporting the significance of the claimed empirical improvements**

Some of the reviewers posed doubts about the competitiveness of our approach with respect to the state-of-the-art (sota). Though our purpose is not to claim sota results, it is still relevant to answer this question. To this end, in the following we present our results in context of the public leaderboard from the WILDS benchmark.


| **Algorithm**  | **PovertyMap** (Worst-U/R r) | **iWildCam** (Macro F1) |
| --- | --- | --- |
| Fish | - | 22.0 (1.8) |
| IRMX (PAIR opt) | 0.47 (0.09) | 27.9 (0.9) |
| IRMX | 0.45 (0.05) | 26.7 (1.1) |
| CORAL | 0.44 (0.07) | 32.7 (0.2) |
| CGD | 0.43 (0.04) | - |
| Group DRO | 0.39 (0.06) | 23.8 (2.0) |
| C-Mixup | **0.53 (0.07)** | - |
| ABSGD | - | 33.0 (0.6) |
| ERM w/ data aug | 0.49 (0.06) | 32.2 (1.2) |
| ERM (more checkpoints) | - | 32.0 (1.5) |
| ERM (grid search) | 0.45 (0.06) | 30.8 (1.3) |
| ERM (rand search) | 0.5 (0.07) | 30.6 (1.1) |
| IRM | 0.43 (0.07) | 15.1 (4.9) |
| ARM-BN | - | 23.3 (2.8) |
| Test-time BN adaptation | - | 13.8 (0.6) |
| MixUp | - | 13.8 (0.8) |
| JTT | - | 11.0 (2.5) |
| rebalancing (ours) | 0.47 (0.10) | **35.5 (0.8)**  |

To ensure fairness, we have included only leaderboard entries that no not deviate from the official benchmark guidelines.

We observe that:
- **on iWildCam our approach achieves a statistically significant improvement over the leading method ABSGD** (p-value of 0.015 based on a Welch t-test);
- **on PovertyMap, our approach is not significantly worse that the the best perfoming method C-Mixup** (p-value of 0.448 that these samples come from the same distribution according to a Welch t-test).

As for ColorMNIST, though there is no public leaderboard, in our paper we show that our method significantly outperforms 14 other methods, which constitutes a reasonably throrough treatment.

*To summarize, we have presented statistical evidence that our approach is competitive or better than previous methods on these datasets.* We hope that this new data, put our empirical results in better perspective of the literature addressing the doubts expressed. We also re-iterate that, despite this good performance, our main point was to support our theory empirically by showing improvement over ERM, which the above results do show.

As for the criticism that our method's performance depends on how the density estimator is employed (e.g., use of dimensionality reduction). We acknowledge that density estimators are imperfect and that tuning them can affect the final result. However, we point out that most modern machine learning methods have hyperparameters and that, thankfully, there is a generally applicable method to select them by using a validation set (as we do here).

---

### Author Response · Authors · 2024-12-03
**Additional results on Waterbirds and MultiNLI**

As reviewer DzLP suggested testing our method on two aditional experiments (Waterbirds and MultiNLI) in the table below we provide the preliminary results for them, when no aditional information (group labels) are used. This also addresses the final concern raised by reviewer B5w2. Due to time constraints we did not perform the hyperparameter tuning for the embedding projection and normalization we did for the rest of the experiments. Instead we used a concervative choice of using UMAP to down-project the embedding space to 64 dimensions, together with normalizing the embeddings by the maximum vector lenght. We exactly followed the experimental setup from [1] with their proposed hyperparameter sweep for ERM.

| **Method** | **Waterbirds (Worst Acc)** | **MultiNLI (Worst Acc)** |
| --- | --- | --- |
| ERM | 85.5 $\pm$ 1.0 | 67.6 $\pm$ 1.2 |
| JTT | 85.6 $\pm$ 0.2 | 67.5 $\pm$ 1.9 |
| RWY | 86.1 $\pm$ 0.7 | 68.0 $\pm$ 1.9 |
| SUBY | 82.4 $\pm$ 1.7 | 64.9 $\pm$ 1.4 |
| Rebalancing (UMAP 64) | 85.4 $\pm$ 0.2 | 69.1 $\pm$ 2.9 |

Consistently with the rest of our experiments, the improvement was significant when density fit was good: the negative log likelihood of the MultiNLI training set was approximately two times smaller than of Waterbirds for the trained density model. Accordingly, in MultiNLI we saw a notable improvement over ERM and other baselines that do not use aditional information repored in [1].

[1] Idrissi, Badr Youbi, et al. "Simple data balancing achieves competitive worst-group-accuracy."

---

### Meta-Review · Area_Chair_R5QX · 2024-12-15

**Metareview:**

This work proposes a new method to enhance the out-of-distribution (OOD) performance of a prediction model in the covariate shift setting, by trying to mimic a uniform covariate distribution in training through the proposed distributionally diverse (DD) risk minimization. The authors pointed out the effect of considering a uniform distribution for worst-case risk minimization, and connected the DD risk to the uniform case. Two practical methods are developed to implement the DD risk minimization.

I agree with reviewers' appreciation on the importance of the topic, the insight contribution, development of the practical methods, and the additional results and differentiation with DRO in the rebuttal phase. Nevertheless, there are also concerns raised. Theoretically, the reviewers are still concerned on the rationale of the DD risk from the proposed risk under uniformity perspective, and the practice of preferring the uniform distribution as a reference. The authors argued that uniformity on manifold can be achieved by density estimation, but I'm not sure if the estimated density is already expressed under the base measure of the manifold instead of that of the ambient space. Practically, the method has quite a few degrees of freedom in the implementation detail, which could have a significant effect on the performance. There are concerns on the solidity of the presented results. Reviewers and I expect the authors to make a systematic and convincing reasoning or investigation for a guideline for each implementation choice (e.g., density estimation).

Given this situation, I tend to recommend a reject for the current version, and encourage the authors to further enhance the paper accordingly.

**Additional Comments On Reviewer Discussion:**

All reviewer replied to authors' rebuttal, and some concerns are well addressed in the rebuttal (e.g., difference from DRO, results on more datasets and more baselines). But no reviewer decided to update the score, due to the common concern on the rationale of the DD risk and some persisting unclarities on the empirical results.

---

### Decision · Program_Chairs · 2025-01-22

Reject